# Tensor Power Methods: Faster and Robust for Arbitrary Order

## Abstract

Tensor decomposition is a fundamental method used in various areas to deal with high-dimensional data. Among the widely recognized techniques for tensor decomposition is the Canonical/Polyadic (CP) decomposition, which breaks down a tensor into a combination of rank-1 components. In this paper, we specifically focus on CP decomposition and present a novel faster robust tensor power method (TPM) for decomposing arbitrary order tensors. Our approach overcomes the limitations of existing methods that are often restricted to lower-order ($\leq 3$) tensors or require strong assumptions about the underlying data structure. By applying the sketching method, we achieve a running time of $\widetilde{O}(n^{p-1})$ per iteration of TPM on a tensor of order $p$ and dimension $n$. Furthermore, we provide a detailed analysis applicable to any $p$-th order tensor, addressing a gap in previous works. Our proposed method offers robustness and efficiency, expanding the applicability of CP decomposition to a broader class of high-dimensional data problems.

## 1 Introduction

In the era of data-driven science and technology, high-dimensional data has become ubiquitous across domains such as computational neuroscience (Bentzur et al., 2022), image processing (Bouveyron et al., 2007), and machine learning (Muja & Lowe, 2014). Higher-order ($> 3$) tensors have become a powerful paradigm for handling this high-dimensional data. Unlike matrices, these higher-order tensors provide a natural framework for representing multi-modal relationships in data, but they can be computationally expensive and challenging to analyze. To address this issue, tensor decomposition is introduced to reduce the dimensionality while preserving the essential structure of the data.

Tensor decomposition has become a fundamental tool in many fields (Kolda & Bader, 2009), including supervised and unsupervised learning (Anandkumar et al., 2014; Janzamin et al., 2015), reinforcement learning (Azizzadenesheli et al., 2016), statistics, and computer vision (Shashua & Hazan, 2005). Moreover, with the rapid outbreak of COVID-19 and the emergence of new variants driven by a large infectious population, recent research has applied tensor models to analyze pandemic data (Dulal et al., 2022) and used tensor decomposition to study gene expression related to COVID-19 (Taguchi & Turki, 2021). Since gene expression is typically highly complex, tensor decomposition can efficiently help researchers uncover connections between various variables, thereby enhancing the understanding of complex systems. This, in turn, may foster advancements in biological and medical research, ultimately benefiting public health.

A well-known decomposition method is the Candecomp/Parafac (CP) decomposition (Harshman, 1970; Carroll & Chang, 1970). In CP decomposition, the input tensor is decomposed into a set of rank-1 components. Although decomposing arbitrary tensors is NP-hard (Hillar & Lim, 2013), it becomes feasible for tensors with linearly independent components by applying a whitening procedure to transform them into orthogonally decomposable tensors. The tensor power method (TPM) is a straightforward and effective technique for decomposing an orthogonal tensor and serves as an extension of the matrix power method. To be more specific, TPM requires calculating the inner product of two vectors: one derived from a rank-1 matrix and the other from a segment of a tensor. This type of inner product can be estimated much more efficiently because sketch vectors have significantly lower dimensions, making it more convenient to compute their inner product. Additionally, sketching can be replaced with sampling to approximate inner products (Song et al., 2016).

When there is no noise in the data, the TPM, through random initialization followed by deflation, can effectively recover the components correctly. However, due to the NP-hard nature of arbitrary tensor decomposition, the perturbation analysis of this method is more complex compared to the matrix case. When large amounts of arbitrary noise are added to an orthogonal tensor, its decomposition becomes intractable. Previous research has demonstrated guaranteed component recovery under bounded noise conditions (Anandkumar et al., 2014), with further improvements outlined in (Anandkumar et al., 2017). More recent work (Wang & Anandkumar, 2016) has further refined the noise requirements.

Since real-world datasets are inherently noisy and high-order, existing methods for CP decomposition face significant challenges when applied to such data. Traditional approaches often rely on restrictive assumptions about tensor structure or are limited to low-order tensors ($\leq 3$), thereby constraining their applicability to many real-world scenarios. Moreover, many of these methods suffer from high computational complexity, making them impractical for large-scale or high-dimensional datasets. These limitations underscore the pressing need for a robust and scalable solution capable of handling tensors of arbitrary orders with efficiency and accuracy.

## 1.1 Our Result

Motivated by these challenges, we propose an algorithm that not only relies on milder assumptions but also is suitable for a broader range of tensor choices. Specifically, we generalize the previous robust TPM algorithm for third-order tensors (Wang & Anandkumar, 2016) to tensors of arbitrary orders. Our proposed algorithm, given any *arbitrary-order* tensor $A \in \mathbb{R}^{n^p}$, outputs the estimated eigenvector/eigenvalue pair along with the deflated tensor. We present our main result as follows:

**Theorem 1.1** (Informal version of Theorem D.2). *There is a robust TPM (Algorithm 1) that takes any $p$-th order and dimension $n$ tensor as input, uses $\widetilde{O}(n^p)$ space and $\widetilde{O}(n^p)$ time in initialization, and in each iteration, it takes $\widetilde{O}(n^{p-1})$ time.*

**Notation.** For any matrix $A \in \mathbb{R}^{n \times k}$, we use $\|A\| := \max_{x \in \mathbb{R}^k \setminus \{0\}^k} \|Ax\|_2/\|x\|_2$ to denote the spectral norm of $A$. We use $\|x\|_2 := (\sum_{i=1}^n x_i^2)^{1/2}$ to denote the $\ell_2$ norm of vector $x$. For two vectors $u \in \mathbb{R}^n$ and $v \in \mathbb{R}^n$, we use $\langle u, v \rangle$ to denote inner product, i.e., $\langle u, v \rangle = \sum_{i=1}^n u_i v_i$.

Let $p \geq 1$ denote some integer. We say $E \in \mathbb{R}^{n \times \cdots \times n}$ (where there are $p$ of $n$), if $E$ is a $p$-th order tensor and every dimension is $n$. For simplicity, we write $E \in \mathbb{R}^{n^p}$. If $p = 1$, then $E$ is a vector. If $p = 2$, then $E \in \mathbb{R}^{n \times n}$ is a matrix. If $p = 3$, then $E \in \mathbb{R}^{n \times n \times n}$ is a 3rd-order tensor. For any two unit vectors $x, y$, we define $\cos \theta(x, y) = \langle x, y \rangle$. For a 3rd-order tensor $E \in \mathbb{R}^{n \times n \times n}$, we have $E(a, b, c) = \sum_{i=1}^n \sum_{j=1}^n \sum_{k=1}^n E_{i,j,k} a_i b_j c_k \in \mathbb{R}^n$, $\|E\| := \max_{x:\|x\|_2=1} |E(x, x, x)|$, $E(I, b, c)_i = \sum_{j=1}^n \sum_{k=1}^n E_{i,j,k} b_j c_k \in \mathbb{R}^n, \forall i \in [n]$, and $E(I, I, c)_{i,j} = \sum_{k=1}^n E_{i,j,k} c_k \in \mathbb{R}^{n \times n}, \forall i, j \in [n] \times [n]$.

The notation of tensor for $p = 3$ can be generalized to any $p$-th order tensor for $p > 3$. For $a, b, c \in \mathbb{R}^n$ and $E = a \otimes b \otimes c \in \mathbb{R}^{n \times n \times n}$, we have $E_{i,j,k} = a_i b_j c_k, \forall i \in [n], j \in [n], k \in [n]$. For $E = a \otimes a \otimes a = a^{\otimes 3} \in \mathbb{R}^{n \times n \times n}$, we have $E_{i,j,k} = a_i a_j a_k, \forall i \in [n], j \in [n], k \in [n]$. For $E = \sum_{i=1}^m u_i^{\otimes 3}$, we have $E(a, b, c) = \sum_{i=1}^m (u_i^{\otimes 3}(a, b, c)) = \sum_{i=1}^m \langle u_i, a \rangle \langle u_i, b \rangle \langle u_i, c \rangle \in \mathbb{R}$.

For $\mu \in \mathbb{R}^d$ and $\Sigma \in \mathbb{R}^{n \times n}$, we use $\mathcal{N}(\mu, \Sigma)$ to denote a Gaussian distribution with mean $\mu$ and covariance $\Sigma$. For $x \sim \mathcal{N}(\mu, \Sigma)$, we denote $x$ as a Gaussian vector.

For all, $a \in \mathbb{R}^n$, we use $\max_{i \in [n]} a_i$ to denote a value $b$ over sets $\{a_1, a_2, \cdots, a_n\}$. For any vector $a \in \mathbb{R}^n$, we use $\arg \max_{i \in [n]} a_i$ to denote the index $j$ such that $a_j = \max_{i \in [n]} a_i$.

**Roadmap.** In Section 2, we present the related work. In Section 3, we introduce the techniques used in this paper. In Section 4, we present our main result. In Section 5, we summarize this paper and provide some future research directions in this field.

## 2 RELATED WORK

**Tensor decomposition.** (Hitchcock, 1927) is the first work that proposed the CP decomposition.Several works have focused on the efficient and fast decomposition of tensors (Tsourakakis, 2010; Phan et al., 2013; Choi & Vishwanathan, 2014; Huang et al., 2013; Kang et al., 2012; Wang et al., 2014; Bhojanapalli & Sanghavi, 2015). Later work (Wang et al., 2015) provided a method based on the random linear sketching technique to enable fast decomposition for orthogonal tensors. (Robeva, 2016) studies the properties of symmetric orthogonally decomposable tensors. (Robeva & Seigal, 2017) incorporate the spectral theory into these orthogonally decomposable tensors. Additionally, (Song et al., 2016) provided another approach to importance sampling, with a faster running time. The canonical polyadic decomposition is a very famous and popular technique of decomposition, which is the CANDECOMP / PARAFAC (CP) decomposition (Song et al., 2016). In CP decomposition, a tensor can be broken down into a combination of rank-1 tensors that add up to it (Harshman, 1970), and this combination is the only possible one up to some minor variations, such as scaling and reordering of the tensors. In other words, there is only one way to decompose the tensor, and any other arrangement of the rank-1 tensors that add up to the same tensor is not possible. This property of tensor decomposition is more restrictive than that of matrices, and it holds for a broader range of tensors. Therefore, tensor decomposition is considered to be more rigid than matrix decomposition. In (Wang et al., 2015), multiple applications, including computational neuroscience, data mining, and statistical learning, of tensor decomposition are mentioned. (Kileel et al., 2021; Wang et al., 2025; Kileel & Pereira, 2025) present a power method for CP decomposition for both symmetric and asymmetric tensors of arbitrary order. The works also guarantees SS-HOPM converges with thorough analysis. In contrast, our work focuses on using sketching techniques to obtain a faster TPM for arbitrary order $p$ in the orthogonal tensor setting

**Unique tensor decomposition.** Previous research in algebraic statistics has already linked tensor decompositions to the development of probabilistic models. By breaking down specific moment tensors using low-rank decompositions, researchers could decide the extent of the identifiability of latent variable models (Allman et al., 2009a;b; Rhodes & Sullivant, 2012). The utilization of Kruskal's theorem in (Kruskal, 1977) was crucial in establishing the accuracy of identifying the model parameters. Nevertheless, this method assumes that people can use an infinite number of samples and cannot provide any information on what is the minimum sample size required to learn the model parameters in these given error bounds. Relying solely on Kruskal's theorem does not suffice to determine the bounds of sample complexity, since by using it, we can only get that the low-rank decompositions of actual moment tensors are unique, but we cannot get enough information about the decomposition of empirical moment tensors. Considering the necessary sample size to learn the parameters of the model, we need to get a uniqueness guarantee which is more robust. We need this guarantee satisfying the requirement that whenever $T'$, which is an empirical moment tensor, closely approximates $T$, which is a moment tensor, a low-rank decomposition of $T'$ would also closely resemble a low-rank decomposition of $T$.

Due to space constraints, we move related works of Canonical/Polydic decomposition, Tucker decomposition and Power Method to Appendix A.1, Sketching techniques to Appendix A.2 and Appendix A.3.

## 3 TECHNIQUE OVERVIEW

In this section, we present a summary of the methods used in our analysis. Since our formal proofs presented in the appendix are very long, we use this section to present the sketch of proofs for the important lemmas and theorems. Specifically, in paragraphs "recoverability of eigenvectors implied by bounded noise" and "analysis of the recoverability", we present the techniques for proving Theorem 4.9. In paragraph "bounding the recovery error", we present the techniques for proving Lemma 3.1 (or equivalently Theorem D.1). Finally, in the paragraph "sketching technique", we present how we use the sketching method to generate the $(1 \pm \epsilon)$ approximation, which supports Lemma 4.2.

**Loosened assumption.** Our main breakthrough is that we generalize the robust tensor power method to support any order tensors. It efficiently resolves the drawback of the earlier method

in (Wang & Anandkumar, 2016) that is limited in the tensor of order below 3 and requires very strict assumptions. Moreover, we have created a strong and adaptable algorithm that can handle a variety of tensor data: natural language corpora, images, videos, etc. Then, we explain how we generalize this in detail.

**Recoverability of eigenvectors implied by bounded noise.** Starting from the construction of the input tensor $A = A^* + \widetilde{E} \in \mathbb{R}^{n^p}$ where it consists of a part of decomposable tensor $A^*$ and a noise term $\widetilde{E}$, we show that, for $u_t \in \mathbb{R}^n$ being a unit vector and $c_0 \geq 1$ and $\epsilon > 0$, if the norm is bounded, in the form of $\|\widetilde{E}(I, u_t, \cdots, u_t)\|_2 \leq 6\epsilon/c_0$ and $|\widetilde{E}(v, u_t, \cdots, u_t)| \leq 6\epsilon/(c_0\sqrt{n})$, where $u_t$ is the approximate eigenvector at iteration $t$ of our algorithm (see Algorithm 1), $v_j$ is one of the orthonormal eigenvectors of the original, unperturbed tensor $A^*$. Then the compositions of $A^*$ is able to be recovered from $A$ (see details in Appendix D). In this paper, we focus on symmetric tensors. The results can be directly extended to asymmetric tensors because these tensors can first be symmetrized using simple matrix operations (Anandkumar et al., 2012). Formally, the eigenvectors have the following properties:

**Part 1.** The difference of the tangent from an eigenvector to the two unit vectors is bounded by a term $18\epsilon/(c_0\lambda_1)$ of the corresponding eigenvalue (see definition of $\tan\theta$ in Def. 4.3):

$$\tan\theta(v_1, u_{t+1}) \leq 0.8\tan\theta(v_1, u_t) + 18\epsilon/(c_0\lambda_1).$$

**Part 2.** Tail components are bounded by the top component, in the power of $p - 2$:

$$\max_{j \in [k]\backslash\{1\}} \lambda_j |v_j^\top u_t|^{p-2} \leq (1/4)\lambda_1 |v_1^\top u_t|^{p-2}.$$

**Part 3.** With all $j$ being an arbitrary element in $\{2, \cdots, k\}$,

$$|v_j^\top u_{t+1}|/|v_1^\top u_{t+1}| \leq 0.8|v_j^\top u_t|/|v_1^\top u_t| + 18\epsilon/(c_0\lambda_1\sqrt{n}).$$

As these are generalized statements from previous results (Wang & Anandkumar, 2016; Anandkumar et al., 2014) from bounded order ($p \leq 3$) to general order $p$, the proof requires a much different analysis. We described the details of our approach in the following paragraph.

**Analysis of the recoverability.** To show part 1 (see the details in Appendix C), we have to find the upper bound of $\tan\theta(v_1, u_{t+1})$. We first turn the tangent into terms of sine and cosine, which can be represented by the norm of the tensors. Then by simply using Cauchy-Schwarz, we can find the upper bound of the term by $\tan\theta(v_1, u_{t+1}) \leq \frac{\|V^\top A^*(I, u_t, \cdots, u_t)\|_2 + \|V^\top \widetilde{E}_{u_t}\|_2}{|v_1^\top A^*(I, u_t, \cdots, u_t)| - |v_1^\top \widetilde{E}_{u_t}|}$ , where $V = (v_2, \cdots, v_k, \cdots, v_n) \in \mathbb{R}^{n \times (n-1)}$ is an orthonormal basis and is the complement of $v_1$. A tensor is said to be orthogonally decomposable if in the above decomposition $\langle v_i, v_j \rangle = 0$ for all $i \neq j$. Using a property for orthogonal tensor that, for $A^* = \sum_{j=1}^k \lambda_j v_j^{\otimes p} \in \mathbb{R}^{n^p}$, it holds that for any $j \in [k]$, $|v_j^\top A^*(I, u, \cdots, u)| = \lambda_j |v_j^\top u|^{p-1}$, we are able to upper bound $\tan\theta(v_1, u_{t+1})$ with $\tan\theta(v_1, u_t)$ in the form of $\tan\theta(v_1, u_{t+1}) \leq \tan\theta(v_1, u_t) \cdot \frac{1}{4} \cdot B_1 + B_1 \cdot B_2$, where $B_1$ and $B_2$ are two simplified terms defined as $B_1 := \frac{1}{1 - |v_1^\top \widetilde{E}_{u_t}|/(\lambda_1 |v_1^\top u_t|^{p-1})}$ and $B_2 := \frac{\|V^\top \widetilde{E}_{u_t}\|_2}{\lambda_1 |v_1^\top u_t|^{p-1}}$. Using the constraint on $\widetilde{E}$ in Theorem 4.9 and Corollary C.12, we further show that $B_1 \leq 1.1$ and $B_2 \leq 18\epsilon/(c_0\lambda_1)$. Combining all these, we complete the proof of the first property.

Regarding the second part, using the property for orthogonal tensor, we lower bound the term $\frac{|v_1^\top u_{t+1}|}{v_j^\top u_{t+1}} \geq \frac{\frac{9}{10}|v_1^\top u_t|}{\frac{1}{4}|v_j^\top u_t| + \frac{1}{10}|v_1^\top u_t|}$. We then divide the proof into two conditions. First, if $|v_j^\top u_t| < |v_1^\top u_t|$, then the proportion of the top component over other rest components can be easily lower bounded by $\frac{\lambda_1 |v_1^\top u_{t+1}|^{p-2}}{\lambda_j |v_j^\top u_{t+1}|^{p-2}} \geq \frac{\lambda_1}{\lambda_j} \cdot 2^{p-2}$. For the opposite condition that $|v_j^\top u_t| \geq |v_1^\top u_t|$, we give a more comprehensive analysis than previous work (see (Wang & Anandkumar, 2016)'s Lemma C.2). We show that for all $p$ being greater than or equal to 3, it holds that $\frac{\lambda_1 |v_1^\top u_{t+1}|^{p-2}}{\lambda_j |v_j^\top u_{t+1}|^{p-2}} \geq 4 \cdot 2^{p-2}$. The final property is also proved in a similar way. For simplicity, we first define two terms $B_3 := \frac{1}{1 - |v_1^\top \widetilde{E}_{u_t}|/(\lambda_1 |v_1^\top u_t|^{p-1})}$ and $B_4 := \frac{|v_j^\top \widetilde{E}_{u_t}|}{\lambda_1 |v_1^\top u_t|^{p-1}}$. Similarly, we find the upper bound

$\frac{|v_j^\top u_{t+1}|}{|v_1^\top u_{t+1}|} \leq \frac{|v_j^\top u_t|}{|v_1^\top u_t|} \cdot \frac{1}{4} B_3 + B_3 \cdot B_4$. $B_3$ can be easily bounded by a similar proof if $|v_1^\top u_t| \geq 1 - \frac{1}{c_0^2 p^2 k^2}$. For $B_4$, we divide it into two case: $|v_1^\top u_t| \leq 1 - \frac{1}{c_0^2 p^2 k^2}$ and $|v_1^\top u_t| > 1 - \frac{1}{c_0^2 p^2 k^2}$. By a different discussion, we can show that $B_4 \leq 18\epsilon/(c_0 \lambda_1 \sqrt{n})$.

**Bounding the recovery error** We now step to the final technical lemma which shows the bound of the approximation error of the output of our algorithm:

**Lemma 3.1** (Informal version of Theorem D.1). *Let $p \geq 3$, $k \geq 1$, and $A = A^* + E \in \mathbb{R}^{n^p}$ be an arbitrary tensor satisfying $A^* = \sum_{i=1}^k \lambda_i v_i^{\otimes p}$. Suppose that $\lambda_1$ is the greatest values in $\{\lambda_i\}_{i=1}^k$ and $\lambda_k$ is the smallest values in $\{\lambda_i\}_{i=1}^k$. The outputs obtained from the robust tensor power method are $\{\widehat{\lambda}_i, \widehat{v}_i\}_{i=1}^k$. Let $E$ satisfy that $\|E\| \leq \epsilon/(c_0 \sqrt{n})$. Then, there exists a permutation $\pi : [k] \to [k]$, such that $\forall i \in [k]$, $|\lambda_i - \widehat{\lambda}_{\pi(i)}| \leq \epsilon$ and $\|v_i - \widehat{v}_{\pi(i)}\|_2 \leq \epsilon/\lambda_i$.*

This Lemma is the key component of our main Theorem (Theorem 1.1). We use mathematical induction to prove this Lemma (Section D). To show the base case, we need to bound three different terms, namely $|\widehat{v}_1 - v_1|$, $|\widehat{\lambda}_1 - \lambda_1|$, and $|\widehat{v}_1^\top v_j|$.

To bound $|\widehat{v}_1 - v_1|$, we need to utilize the properties of angle and apply the definitions and Lemmas we develop in Section 4. First, we can show $\tan \theta(u_0, v_1) \leq \sqrt{n}$. By using the fact that $|u_{t^*}^\top v_1| = 1 - \frac{1}{c_0^2 p^2 k^2}$ together with some respective properties of $u_{t^*}^\top$ and $v_1$, we can get $\|u_{t^*} - v_1\|_2^2 = 2/(c_0^2 p^2 k^2)$. Finally, we can bound $|\widehat{v}_1 - v_1|$ using this information and recursively applying Part 1 of Theorem 4.9.

For the second term $|\widehat{\lambda}_1 - \lambda_1|$, we simplify it and split that into three parts, namely $B_5$, $B_6$, and $B_7$ which are defined as follows

- $B_5 := |\widetilde{E}(\widehat{v}_1, \cdots, \widehat{v}_1)|$,

- $B_6 := |\lambda_1 |v_1^\top \widehat{v}_1|^p - \lambda_1|$, and

- $B_7 := \sum_{j=2}^k \lambda_j |v_j^\top \widehat{v}_1|^p$..

It suffices to bound these three terms. Using the properties of tensor spectral norm and various inequalities we develop in Section D, we prove that $B_5 \leq \epsilon/12$, $B_6 \leq \epsilon/12$, and $B_7 \leq \epsilon/4$. By putting these together, we get that $|\widehat{\lambda}_1 - \lambda_1| \leq \epsilon/12 + \epsilon/12 + \epsilon/4 \leq \epsilon$. Moreover, we need to give $\epsilon$ a proper value. If $\epsilon$ is too big, we might not get our desired result. On the other hand, if $\epsilon$ is too small, the result might be meaningless. Finally, by setting $\epsilon < \frac{1}{4} k^{1/(p-1)} \lambda_k$, we get the desired result.

What is left out is the third term $|\widehat{v}_1^\top v_j|$. We need to recursively apply the third part of Theorem 4.9. We show that $|v_j^\top u_{t^*}|/|v_1^\top u_{t^*}| \leq 0.8^{t^*} \cdot 1/(1/\sqrt{n})$. In the end, by choosing proper $T$ and $t^*$ values, we can get our desired bound.

In the inductive case, the arrangement of the proof is just like the ones in the base case: we also need to bound these three terms. Moreover, for $i$ being larger, we also need to consider the noise, namely $\widetilde{E} = E + \sum_{i=1}^r E_i + \overline{E} \in \mathbb{R}^{n^p}$, which adds more complexity to the condition we encounter.

**Sketching technique.** Inspired by a recent sketching technique (Cherapanamjeri & Nelson, 2020), we apply a similar sketching operation to develop a distance estimation data structure to apply in our tensor power method. Our data structure uses the Randomized Hadamard Transform (RHT) to generate the sketching matrix. The data structure stores the sketches of a set of maintained tensors $\{A_i\}_{i\in[n]} \subseteq \mathbb{R}^{n^{p-1}}$. Let $A^* = \sum_{i=1}^k \lambda_i v_i^{\otimes p}$, then $A_i$ is the order-$(p-1)$ slice of $A^*$, i.e., $\sum_{j=1}^k \alpha_{i,j} x_j^{\otimes(p-1)}$. Now, when a query tensor of the form $q = u^{\otimes(p-1)}$ comes, our data structure can read $\{x_j\}_{j\in[k]}, \alpha \in \mathbb{R}^{n \times k}, u \in \mathbb{R}^n$, and return an $(1 \pm \epsilon)$ estimated product $v \in \mathbb{R}^n$ such it approximates $\langle A_i - \sum_{j=1}^k \alpha_{i,j} x_j^{\otimes(p-1)}, u^{\otimes(p-1)} \rangle$. This procedure runs fast in time $\widetilde{O}(\epsilon^{-2} n^{p-1} + n^2 k)$. Applying this data structure when computing the error, we are able to achieve our final fast TPM algorithm.

# 4 ROBUST TENSOR POWER METHOD ANALYSIS FOR GENERAL ORDER $p \geq 3$

The goal of this section is to give a sketch of the proof of our main result (see Theorem 1.1). Comparing with Section 3, which present the techniques for proving the important components of our main result, namely Lemma 3.1 and Theorem 4.9, in this section, we move on to the high level picture where how these important components may support Theorem 1.1 and Algorithm 1. In Section 4.1, we give an overview of our main algorithm and present the meaning of the important data structures being used in this algorithm, where this main algorithm is paired with our main theorem, Theorem 1.1. In Section 4.2, we analyze the properties of the $p$-th order tensor, where $p$ is an arbitrary positive integer greater than or equal to 3. These properties are generalized from the third and the fourth order tensors. In Section 4.3, we generalize the properties of the existing robust tensor power method from the third order to any arbitrary order greater than or equal to three.

In short, our main theorem can be proved by combining the efficient implementation of the key operations needed in the tensor power method (Lemma 4.2) and the theoretical guarantees for the robust tensor power method (Lemma 3.1).

## 4.1 AN OVERVIEW OF OUR MAIN ALGORITHM

---
**Algorithm 1** Our main algorithm

1: **procedure** FASTTENSOR($A$)
2:     ds.INIT($A$)
3:     **for** $\ell = 1 \to L$ **do**
4:         **for** $t = 1 \to T$ **do**
5:             $u^{(\ell)} \leftarrow$ ds.QUERY($u^{(\ell)}$)               ▷ Lemma 4.2
6:             $u^{(\ell)} \leftarrow u^{(\ell)}/\|u^{(\ell)}\|_2$
7:         **end for**
8:         $\lambda^{(\ell)} \leftarrow$ ds.QUERYVALUE($u^{(\ell)}$)         ▷ Lemma 4.2
9:     **end for**
10:    $\ell^* \leftarrow \arg\max_{\ell \in [L]} \lambda^{(\ell)}$
11:    $u^* \leftarrow u^{(\ell^*)}$
12:    **for** $t = 1 \to T$ **do**
13:        $u^* \leftarrow$ ds.QUERY($u^*$)
14:        $u^* \leftarrow u^*/\|u^*\|_2$
15:    **end for**
16:    $\lambda^* \leftarrow$ ds.QUERYVALUE($u^*$)
17:    **return** $\lambda^*, u^*$
18: **end procedure**

---

In our main algorithm (Algorithm 1), we use ds.INIT($A$) to initialize the data structure. INIT can take $n$ tensors, $A_1, A_2, A_3, \ldots, A_n \in \mathbb{R}^{n^{p-1}}$. We use ds.QUERY($u^{(\ell)}$), which takes $u^{(\ell)} \in \mathbb{R}^n$ as an input, to output a vector $v^\ell \in \mathbb{R}^n$, where each entry of $v^{(\ell)}$ is an approximation of $\langle A_i, u^{\otimes(p-1)} \rangle$, for all $i \in [n]$. Finally, ds.QUERYVALUE($u^{(\ell)}$) is similar to ds.QUERY($u^{(\ell)}$): it takes $u^{(\ell)} \in \mathbb{R}^n$ as an input and output a real number $\lambda^{(\ell)} \in \mathbb{R}$, which is an approximation of $\langle A, u^{\otimes p} \rangle$.

Below, we present the efficient implementation of the data structure we need.

**Definition 4.1** (Finding the top eigenvector and top-$k$ eigenvectors). *Given a collection of $n$ tensors $A_1, A_2, \cdots, A_n \in \mathbb{R}^{n^{p-1}}$, the goal is to design a structure that supports the following operations*

- INIT $(A_1, \cdots, A_n \in \mathbb{R}^{n^{p-1}})$. *It takes $n$ tensors as inputs and creates a data structure.*

- QUERY $(u \in \mathbb{R}^n)$, *the goal is to output a vector $v \in \mathbb{R}^n$ such that $v_i \approx \langle A_i, u^{\otimes(p-1)} \rangle$, $\forall i \in [n]$*

- QUERY($\{x_i\}_{i \in [k]} \in \mathbb{R}^n, \alpha \in \mathbb{R}^{n \times k}, u \in \mathbb{R}^n$). *the goal is to output a vector $v \in \mathbb{R}^n$ such that $v_i \approx \langle A_i - \sum_{j=1}^k \alpha_{i,j} x_j^{\otimes(p-1)}, u^{\otimes(p-1)} \rangle$, $\forall i \in [n]$*

We state our data structure as follows:

**Lemma 4.2** (Data Structure). *Given $n$ tensors $A_1, A_2, \cdots A_n \in \mathbb{R}^{n^{p-1}}$ where $\|A_i\|_F \leq D_i, \forall i \in [n]$, we let $\|A\|_F \leq D$. Let $\epsilon, \delta \in (0, 1/2)$. Then, there exists a randomized data structure with the following operations:*

- INIT($A_1, \cdots, A_n \in \mathbb{R}^{n^{p-1}}$): *It preprocesses $n$ tensors, in time $\widetilde{O}(\epsilon^{-2} n^p \log(1/\delta))$.*

- QUERY($u \in \mathbb{R}^n$). *It takes a unit vector $u \in \mathbb{R}^n$ as input. The goal is to output a vector $v \in \mathbb{R}^n$ such that for all $i \in [n]$, $(1 - \epsilon) \cdot \langle A_i, u^{\otimes(p-1)} \rangle - D_i \cdot \epsilon \leq v_i \leq (1 + \epsilon) \cdot \langle A_i, u^{\otimes(p-1)} \rangle + D_i \cdot \epsilon$. This can be done in time $\widetilde{O}(\epsilon^{-2} n^{(p-1)} \log(1/\delta))$.*

- QUERYVALUE($u \in \mathbb{R}^n$). *The goal is to output a number $v \in \mathbb{R}$ such that $(1-\epsilon)\langle A, u^{\otimes p} \rangle - D \cdot \epsilon \leq v \leq (1 + \epsilon)\langle A, u^{\otimes p} \rangle + D \cdot \epsilon$. This can be done in time $\widetilde{O}(\epsilon^{-2} n^{(p-1)} \log(1/\delta))$.*

- QUERYRES($\{x_j\}_{j \in [k]} \in \mathbb{R}^n, \alpha \in \mathbb{R}^{n \times k}, u \in \mathbb{R}^n$). *The goal is to output a vector $v \in \mathbb{R}^n$ such that for all $i \in [n]$,*

$$
(1 - \epsilon) \cdot \langle A_i - \sum_{j=1}^k \alpha_{i,j} x_j^{\otimes(p-1)}, u^{\otimes(p-1)} \rangle - D_i \cdot \epsilon \leq v_i
$$

$$
\leq (1 + \epsilon) \cdot \langle A_i - \sum_{j=1}^k \alpha_{i,j} x_j^{\otimes(p-1)}, u^{\otimes(p-1)} \rangle + D_i \cdot \epsilon.
$$

*This can be done in time $\widetilde{O}(\epsilon^{-2} n^{(p-1)} \log(1/\delta) + n^2 k)$.*

*All the queries are robust to adversary type queries.*

*Proof.* The correctness of INIT and QUERY directly follows from (Cherapanamjeri & Nelson, 2020).

For the QUERYRESIDUAL, the running time only need to pay an extra term is computing $\langle \sum_{j=1}^k \alpha_{i,j} x_j^{\otimes(p-1)}, u^{\otimes(p-1)} \rangle$ which is sufficient just to compute $\sum_{j=1}^k \alpha_{i,j} \langle x_j, u \rangle^{p-1}$. The above step takes $O(kn)$ time. Since there are $n$ different indices $i$. So overall extra time is $O(n^2 k)$. □

### 4.2 USEFUL FACTS

We finish presenting the efficient implementation of the key operations. Now, we move on to the sketch of proof for the theoretical guarantees for the robust tensor power method (Lemma 3.1). Proving this is not trivial, as we presented in the technique overview (see Section 3). We need to first prove some important facts, where these facts are frequently used in the proof of Theorem 4.9, and then generalize Theorem 4.9 to obtain Lemma 3.1. First, we give the formal definitions of sin, cos, and tan.

**Definition 4.3.** *For $u, v$ be unit vectors, we define $\cos \theta(u, v) := \langle u, v \rangle$, $\sin \theta(u, v) := \sqrt{1 - \cos^2 \theta(u, v)}$ and $\tan \theta(u, v) := \sin \theta(u, v) / \cos \theta(u, v)$.*

We use the following facts to support the analysis of recoverability.

**Fact 4.4** (Informal version of Fact B.7). *Let $p \geq 3$. Let $A^* = \sum_{j=1}^k \lambda_j v_j^{\otimes p} \in \mathbb{R}^{n^p}$ be the orthogonal tensor. Then, for all $j \in [k]$, given a vector $u \in \mathbb{R}^n$, we can get $|v_j^\top A^*(I, u, \cdots, u)| = \lambda_j |v_j^\top u|^{p-1}$.*

The following fact provides the upper bound for $E(u, v, \cdots, v)$ and $\|E(I, v, \ldots, v)\|_2$, which is used for the norm bounding analysis (see details in Section C and D).

**Fact 4.5.** *Let $E \in \mathbb{R}^{n^p}$ is an arbitrary orthogonal tensor and $u, v \in \mathbb{R}^n$ are two arbitrary unit vectors. Then, we have $|E(u, v, \cdots, v)| \leq \|E\|$ and $\|E(I, v, \ldots, v)\|_2 \leq \sqrt{n} \|E\|$.*

*Proof.* Part 1 follows trivially from the definition of $\|E\|$.

For part 2, we define a unit vector $w \in \mathbb{R}^n$ to be $(1/\sqrt{n}, \cdots, 1/\sqrt{n})$,

$$
\begin{aligned}
\|E(I, v, \ldots, v)\|_2^2 &= \sum_{i_1=1}^{n} \left( \sum_{i_2=1}^{n} \cdots \sum_{i_p=1}^{n} E_{i_1, i_2, \cdots, i_p} v_{i_2} \cdots v_{i_p} \right)^2 \\
&= n \sum_{i_1=1}^{n} \left( \sum_{i_2=1}^{n} \cdots \sum_{i_p=1}^{n} E_{i_1, i_2, \cdots, i_p} w_{i_1} v_{i_2} \cdots v_{i_p} \right)^2 \\
&\leq n \|E\|^2,
\end{aligned}
$$

where the first step follows from the definition of $E(I, v, \ldots, v)$, the second step follows from our definition for $w$, and the last step follows from $n \geq 1$. This result implies $\|E(I, v, \cdots, v)\|_2 \leq \sqrt{n} \|E\|$. $\qquad \square$

**Fact 4.6** (Informal version of Fact B.8). *Let $p$ is greater than or equal to 3, $x, y, u, v \in \mathbb{R}^n$ be any arbitrary unit vectors, and $j \in \{0, 1, \cdots, p-2\}$. Then, we have*

$$
\|[x \otimes v^{\otimes(p-1)}](I, u, \cdots, u) - [y \otimes v^{\otimes(p-1)}](I, u, \cdots, u)\|_2 = |\langle u, v \rangle|^{p-1} \cdot \|x - y\|_2 \quad (1)
$$

*and*

$$
\begin{aligned}
&\|[v^{\otimes(1+j)} \otimes x \otimes v^{\otimes(p-2-j)}](I, u, \cdots, u) - [v^{\otimes(1+j)} \otimes y \otimes v^{\otimes(p-2-j)}](I, u, \cdots, u)\|_2 \\
&\leq |\langle u, v \rangle|^{p-2} \cdot \|x - y\|_2.
\end{aligned} \quad (2)
$$

The following fact transforms the $\ell_2$ norm into the form of the sum of a list of real numbers, which helps us with simplifying $\|V^\top A^*(I, u, \cdots, u)\|_2^2$ to support the analysis of the recoverability (see Section C for details).

**Fact 4.7.** *Let $v_1, v_2, \cdots, v_n$ be an orthonormal basis. Let $V = (v_2, \cdots, v_n) \in \mathbb{R}^{n \times (n-1)}$. Let $A^* = \sum_{i=1}^{k} \lambda_i v_i^{\otimes p}$. Let $u \in \mathbb{R}^n$ be a vector. Then, we have $\|V^\top A^*(I, u, \cdots, u)\|_2^2 = \sum_{j=2}^{k} \lambda_j^2 |v_j^\top u|^{2(p-1)}$.*

*Proof.* We have

$$
\|V^\top A^*(I, u, \cdots, u)\|_2^2 = \sum_{j=2}^{k} |v_j^\top A^*(I, u, \cdots, u)|^2 = \sum_{j=2}^{k} (\lambda_j |v_j^\top u|^{p-1})^2 = \sum_{j=2}^{k} \lambda_j^2 |v_j^\top u|^{2(p-1)},
$$

where the first step follows from the definition of $\ell_2$ norm, the second step follows from Fact B.7, and the last step follows from simple algebra. $\qquad \square$

### 4.3 Convergence guarantee and deflation

Consequently, in this section, with the help of these technical facts, we are ready to present the second component necessary to support our main theorem (Theorem 1.1), specifically Lemma 3.1. We generalize the robust tensor power method to all cases where $p \geq 3$.

**Lemma 4.8.** *Let $t \in [k]$. Let $\eta \in (0, 1/2)$. In $\mathbb{R}^n$, $\mathcal{U}$ represents a set of random Gaussian vectors. Let $|\mathcal{U}| = \Omega(k \log(1/\eta))$. Then, there is a probability of at least $1 - \eta$ that there exists a vector $u \in \mathcal{U}$ satisfying the following condition: $\max_{j \in [k] \setminus \{t\}} |v_j^\top u| \leq \frac{1}{4} |v_t^\top u|$ and $|v_t^\top u| \geq 1/\sqrt{n}$.*

We analyze (Wang & Anandkumar, 2016)'s Lemma C.2 and generalize it from $p$ being equal to 3 to any $p$ being greater than or equal to 3.

In the following Theorem, intuitively, we treat $A^*$ as the ground-truth tensor. We treat $\widetilde{E}$ as the noise tensor. In reality, we can not access the $A^*$ directly. We can only access $A^*$ with some noise which is $\widetilde{E}$. But whenever $\widetilde{E}$ (the noise) is small compared to ground-truth $A^*$, then we should be able to recover $A^*$.

**Theorem 4.9.** *Let $\widetilde{E} \in \mathbb{R}^{n^p}$ denote some tensor representing the noise. Let $c > 0$ is an arbitrarily small number and $c_0 \geq 1$. Let $p$ be greater than or equal to 3. $A = A^* + \widetilde{E} \in \mathbb{R}^{n^p}$ is an arbitrary tensor satisfying $A^* = \sum_{i=1}^k \lambda_i v_i^{\otimes p}$, where $A^*$ is orthogonal decomposable.*

*Let*

$$u_{t+1} = \frac{A(I, u_t, \cdots, u_t)}{\|A(I, u_t, \cdots, u_t)\|_2},$$

*where $u_t \in \mathbb{R}^n$ is an unit vector.*

*We define Event $\xi$ to be*

$$|v_1^\top u_t| \leq 1 - 1/(c_0^2 p^2 k^2).$$

*Let $0 < \epsilon \leq \frac{c\lambda_1}{(c_0 p^2 k n^{(p-2)/2})}$. Let $T = \Omega(\log(\lambda_1 n/\epsilon))$. Let $t \in [T]$.*

*Suppose*

$$\|\widetilde{E}(I, u_t, \cdots, u_t)\|_2 \leq \begin{cases} 4p\epsilon, & \text{if } \xi \\ 6\epsilon/c_0, & \text{ow.} \end{cases}$$

*and $|\widetilde{E}(v, u_t, \cdots, u_t)| \leq \begin{cases} 4\epsilon/\sqrt{n} & \text{if } \xi \\ 6\epsilon/(c_0\sqrt{n}) & \text{ow} \end{cases}$*

*Then,*

    *1. We have*

$$\tan\theta(v_1, u_{t+1}) \leq \begin{cases} 0.8 \tan\theta(v_1, u_t) & \text{if } \xi \\ 0.8 \tan\theta(v_1, u_t) + 18\frac{\epsilon}{c_0\lambda_1} & \text{ow} \end{cases} \tag{3}$$

    *2. We have*

$$\max_{j \in [k]\setminus\{1\}} \lambda_j |v_j^\top u_t|^{p-2} \leq (1/4)\lambda_1 |v_1^\top u_t|^{p-2}. \tag{4}$$

    *3. For any $j \in \{2, \cdots, k\}$, we have*

$$\frac{|v_j^\top u_{t+1}|}{|v_1^\top u_{t+1}|} \leq \begin{cases} 0.8|v_j^\top u_t|/|v_1^\top u_t| & \text{if } \xi \\ 0.8|v_j^\top u_t|/|v_1^\top u_t| + 18\epsilon/(c_0\lambda_1\sqrt{n}) & \text{ow} \end{cases} \tag{5}$$

Because of the space limit, the formal proof is deferred to Appendix C. Theorem 4.9 provides key properties of the tensor power method for a single iteration. It shows how the algorithm converges towards the dominant eigenvector and how errors are controlled in each step. Finally, using Theorem 4.9, we can prove Lemma 3.1 that our algorithm recovers the tensor components (eigenvectors and eigenvalues) up to a specified error bound using mathematical induction. Combining this with our fast sketching technique (Lemma 4.2), we finally prove our main Theorem (Theorem 1.1).

## 5 CONCLUSION

We present a robust tensor power method that supports arbitrary order tensors. Our method overcomes the limitations of existing approaches, which are often restricted to lower-order tensors or require strong assumptions about the underlying data structure. This requires non-trivial mathematical tools to handle the added complexity. We develop new properties of higher-order tensors and analyze the convergence and error bounds. By leveraging advanced techniques from optimization and linear algebra, we have developed a powerful and flexible algorithm that can handle a wide range of tensor data, from images and videos to multivariate time series and natural language corpora. We believe that our result has some insights into various tasks, including tensor decomposition, low-rank tensor approximation, and independent component analysis. We believe that our contribution will significantly advance the field of tensor analysis and provide new opportunities for handling high-dimensional data in various domains. We here propose some future directions. We encourage extending our method to more challenging scenarios, such as noisy data analysis, and exploring its applications in emerging areas, such as neural networks and machine learning.

## ETHIC STATEMENT

This paper does not involve human subjects, personally identifiable data, or sensitive applications. We do not foresee direct ethical risks. We follow the ICLR Code of Ethics and affirm that all aspects of this research comply with the principles of fairness, transparency, and integrity.

## REPRODUCIBILITY STATEMENT

We ensure reproducibility of our theoretical results by including all formal assumptions, definitions, and complete proofs in the appendix. The main text states each theorem clearly and refers to the detailed proofs. No external data or software is required.

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

# Appendix

**Roadmap.** In Section A, we present our additional related works. In Section B, we introduce the background concepts (definitions and properties) that we use in the Appendix. In Section C, we provide more details and explanations to support the properties we developed in this paper. In Section D, we present our important Theorems (Theorem D.1 and Theorem D.2) and their proofs.

## A   ADDITIONAL RELATED WORKS

In Section A.1, we introduce Canonical/Polydic decomposition and Tucker decomposition. In Section A.2, we present some sketching techniques. In Section A.3, we show previous works about power method.

### A.1   CANONICAL/POLYDIC DECOMPOSITION AND TUCKER DECOMPOSITION

The most commonly employed techniques for breaking down tensors are CP (Canonical/Polydic) decomposition and Tucker factorization. CP decomposes a tensor that has higher order into a collection of fixed-rank individual tensors that are summed together, while Tucker factorization reduces a tensor that has higher order to a smaller core tensor and a matrix product of each of its modes. Non-negative tensor factorization is the extension of non-negative matrix factorization to multiple dimensions (Bhatt et al., 2021). Recent research in Tucker decomposition has focused on developing more efficient algorithms for computing the decomposition (Zhou et al., 2015; Kim & Candan, 2016; Fahrbach et al., 2022), improving its accuracy and robustness (Zhang & Ding, 2013; Heng et al., 2022), and applying it to various new domains, such as image representation (Zhang & Ding, 2013).

### A.2   SKETCHING TECHNIQUES

Sketching methods have emerged as a powerful paradigm in numerical linear algebra, serving as a fundamental approach to dimension reduction while preserving essential mathematical properties. These techniques, which originated from the theoretical computer science community, provide a way to project high-dimensional data into lower-dimensional spaces while maintaining important structural information and computational guarantees. They have become increasingly important in machine learning, data science, and scientific computing due to their ability to reduce computational complexity while maintaining accuracy guarantees.

It has played an important role in tensor approximation (Song et al., 2019; Mahankali et al., 2022; Deng et al., 2023), matrix completion (Gu et al., 2023), submodular function maximization (Qin et al., 2023), dynamic sparsifier (Deng et al., 2022a), dynamic tensor produce regression (Reddy et al., 2022), semi-definite programming (Song et al., 2022b), sparsification problems involving an iterative process (Song et al., 2022a), adversarial training (Gao et al., 2022), kernel density estimation (Qin et al., 2022), and distance oracle problem (Deng et al., 2022b).

### A.3   POWER METHOD

The power method is a popular iterative algorithm for computing the dominant eigenvector and eigenvalue of a tensor. In recent years, there is a series of works (Chang et al., 2008; Ng et al., 2010; Wang et al., 2009) that focused on this topic. The work of (Kolda & Mayo, 2011) provides the result to compute real symmetric-tensor eigenpairs, which is closely related to the optimal rank-1 approximation of a symmetric tensor. Moreover, their method is based on the shifted symmetric higher-order power method (SS-HOPM), which can be viewed as a generalization of the power iteration method for matrices. (Anandkumar et al., 2014) considers the relation between tensor decomposition and learning latent variable models, where they also provide a detailed analysis of a robust TPM. More recent work by (Anandkumar et al., 2017) offers a new approach to analyzing the behavior of tensor power iterations in the overcomplete scenario, in which the tensor's CP rank surpasses the input dimension.

## B PRELIMINARY

In Section B.1, we define several basic notations. In Section B.2, we state several basic facts. In Section B.3, we present facts and tools for tensors.

### B.1 NOTATIONS

In this section, we start to introduce the fundamental concepts we use.

For any function $f$, we use $\widetilde{O}(f)$ to denote $f \cdot \mathrm{poly}(\log f)$.

$\mathbb{R}$ denotes the set that contains all real numbers.

For a scalar $a$, i.e. $a \in \mathbb{R}$, $|a|$ represents the absolute value of $a$.

For any $A \in \mathbb{R}^{n \times k}$ being a matrix and $x \in \mathbb{R}^k$ being a vector, we use $\|A\| := \max_{x \in \mathbb{R}^k} \|Ax\|_2 / \|x\|_2$ to denote the spectral norm of $A$.

We use $\|x\|_2 := (\sum_{i=1}^n x_i^2)^{1/2}$ to denote the $\ell_2$ norm of the vector $x$.

For two vectors $u \in \mathbb{R}^n$ and $v \in \mathbb{R}^n$, we use $\langle u, v \rangle$ to denote the inner product, i.e. $\langle u, v \rangle = \sum_{i=1}^n u_i v_i$.

Let $p \geq 1$ denote some integer. We say $E \in \mathbb{R}^{n \times \cdots \times n}$ (where there are $p$ of $n$), if $E$ is a $p$-th order tensor and every dimension is $n$. For simplicity, we write $E \in \mathbb{R}^{n^p}$. If $p = 1$, then $E$ is a vector. If $p = 2$, $E \in \mathbb{R}^{n \times n}$ is a matrix. If $p = 3$, then $E \in \mathbb{R}^{n \times n \times n}$ is a 3rd order tensor.

For any two vectors $x, y$, we define $\theta(x, y)$ to be $\cos \theta(x, y) = \langle x, y \rangle$.

For a 3rd tensor $E \in \mathbb{R}^{n \times n \times n}$, we have $E(a, b, c) \in \mathbb{R}$

$$E(a, b, c) = \sum_{i=1}^n \sum_{j=1}^n \sum_{k=1}^n E_{i,j,k} a_i b_j c_k.$$

Similarly, the definition can be generalized to $p$-th order tensor.

For a 3rd order tensor $E \in \mathbb{R}^{n \times n \times n}$, we have $E(I, b, c) \in \mathbb{R}^n$,

$$E(I, b, c)_i = \sum_{j=1}^n \sum_{k=1}^n E_{i,j,k} b_j c_k, \quad \forall i \in [n]$$

For a 3rd order tensor $E \in \mathbb{R}^{n \times n \times n}$, we have $E(I, I, c) \in \mathbb{R}^{n \times n}$

$$E(I, I, c)_{i,j} = \sum_{k=1}^n E_{i,j,k} c_k, \quad \forall i, j \in [n] \times [n].$$

Let $a, b, c \in \mathbb{R}^n$. Let $E = a \otimes b \otimes c \in \mathbb{R}^{n \times n \times n}$. We have
$$E_{i,j,k} = a_i b_j c_k, \quad \forall i \in [n], \forall j \in [n], k \in [n]$$

Let $a \in \mathbb{R}^n$, let $E = a \otimes a \otimes a = a^{\otimes 3} \in \mathbb{R}^{n \times n \times n}$. We have
$$E_{i,j,k} = a_i a_j a_k, \quad \forall i \in [n], \forall j \in [n], k \in [n]$$

Let $E = \sum_{i=1}^m u_i^{\otimes 3}$. Then we have $E(a, b, c) \in \mathbb{R}$

$$E(a, b, c) = \sum_{i=1}^m (u_i^{\otimes 3}(a, b, c)) = \sum_{i=1}^m \langle u_i, a \rangle \langle u_i, b \rangle \langle u_i, c \rangle.$$

For $\mu \in \mathbb{R}^d$ and $\Sigma \in \mathbb{R}^{n \times n}$. We use $\mathcal{N}(\mu, \Sigma)$ to denote a Gaussian distribution with mean $\mu$ and covariance $\Sigma$. For $x \sim \mathcal{N}(\mu, \Sigma)$, we denote $x$ as a Gaussian vector.

For any vector $a \in \mathbb{R}^n$, we use $\max_{i \in [n]} a_i$ to denote a value $b$ over sets $\{a_1, a_2, \cdots, a_n\}$.

For any vector $a \in \mathbb{R}^n$, we use $\arg \max_{i \in [n]} a_i$ to denote the index $j$ such that $a_j = \max_{i \in [n]} a_i$.

Let $\mathbb{N}$ denote non-negative integers.

## B.2 BASIC FACTS

In this section, we introduce some basic facts.

**Fact B.1.** *We have*

- *Part 1. For any $x \in (0, 1)$ and integer $p \geq 1$, we have $|1 - (1 - x)^p| \leq p \cdot x$.*

- *Part 2. $(a + b)^p \leq 2^{p-1}a^p + 2^{p-1}b^p$.*

**Fact B.2** (Geometric series). *If the following conditions hold*

- *Let $a \in \mathbb{R}$.*

- *Let $k \in \mathbb{N}$.*

- *Let $r \in \mathbb{R}$ and $0 < r < 1$.*

*Then, for all $k$, the series which can be expressed in the form of*

$$\sum_{i=0}^{k} ar^i$$

*is called the geometric series.*

*Let $a_0$ denote the value of this series when $k = 0$, namely $a_0 = ar^0 = a$.*

*This series is equal to*

    *1.*

$$S_k = \sum_{i=0}^{k} ar^i = a_0 \frac{(1 - r^n)}{1 - r},$$

    *when $k \neq \infty$, or*

    *2.*

$$S_k = \sum_{i=0}^{k} ar^i = \frac{a_0}{1 - r},$$

    *when $k = \infty$.*

**Fact B.3.** *If the following conditions hold*

- *Let $\sum_{n=1}^{\infty} b_n$ be a series.*

- *Let $k \in \mathbb{N}$.*

- *Let $a \in \mathbb{R}$.*

- *Let $r \in \mathbb{R}$ and $0 < r < 1$.*

- *Let $\sum_{i=0}^{k} ar^i$ be a geometric series.*

- *Suppose $\sum_{n=1}^{\infty} b_n \leq \sum_{i=0}^{k} ar^i$.*

*Then, $\sum_{n=1}^{\infty} b_n$ is convergent and is bounded by*

$$\frac{a_0}{1 - r}.$$

*Proof.* By Fact B.2, we get that the geometric series is convergent, for all $k \in \mathbb{N}$.

Then, $\sum_{n=1}^{\infty} b_n$ is convergent by the comparison test.

We have

$$a_0 \frac{(1 - r^n)}{1 - r} \le \frac{a_0}{1 - r} \tag{6}$$

because for all $0 < r < 1$, we have $(1 - r^n) < 1$.

Therefore, we get

$$\sum_{n=1}^{\infty} b_n \le \sum_{i=0}^{k} a r^i$$
$$\le \frac{a_0}{1 - r},$$

where the first step follows from the assumption in the Fact statement and the second step follows from Eq. (6).

$\square$

**Fact B.4.** *If the following conditions hold*

- *$u, v, w \in \mathbb{R}^n$ are three arbitrary unit vectors.*

- *For all $x$ satisfying $0 \le x \le 1$.*

- *Suppose $1 - x \le \langle u, w \rangle$.*

- *Suppose $\langle v, w \rangle = 0$.*

*Then $\langle u, v \rangle \le \sqrt{2x - x^2}$.*

*Proof.* First, we want to show that

$$|\sin \theta(u, w)| = \sqrt{1 - \cos^2 \theta(u, w)}$$
$$= \sqrt{1 - \langle u, w \rangle^2}$$
$$\le \sqrt{1 - (1 - x)^2}, \tag{7}$$

where the first step follows from the definition of $\sin \theta(u, w)$ (see Definition 4.3), the second step follows from the definition of $\cos \theta(u, w)$ (see Definition 4.3), and the last step follows from the assumption of this fact.

Then, we have

$$\langle u, v \rangle = \cos \theta(u, v)$$
$$= |\cos \theta(u, w) \cos \theta(v, w) - \sin \theta(u, w) \sin \theta(v, w)|$$
$$\le |\cos \theta(u, w) \cos \theta(v, w) + \sin \theta(u, w) \sin \theta(v, w)|$$
$$\le |\cos \theta(u, w) \cos \theta(v, w)| + |\sin \theta(u, w) \sin \theta(v, w)|$$
$$= 0 + |\sin \theta(u, w) \sin \theta(v, w)|$$
$$\le |\sin \theta(u, w)| \cdot |\sin \theta(v, w)|$$
$$\le |\sin \theta(u, w)|$$
$$\le \sqrt{1 - (1 - x)^2}$$
$$= \sqrt{2x - x^2},$$

where the first step follows from the definition of $\cos \theta(u, v)$ (see Definition 4.3), the second step follows from $\cos(a + b) = \cos(a) \cos(b) - \sin(a) \sin(b)$, the third step follows from simple algebra, the fourth step follows from the triangle inequality, the fifth step follows from $\cos \theta(v, w) = 0$, the sixth step follows from the Cauchy–Schwarz inequality, the seventh step follows from $|\sin \theta(w, v)| \le 1$, the eighth step follows from Eq. (7), and the last step follows from simple algebra. $\square$

**Fact B.5.** *If the following conditions hold*

- *Let $E \in \mathbb{R}^{n^p}$.*

- *Let $u, v \in \mathbb{R}^n$ be two vectors.*

*Then*

- $|E(v, u, \cdots, u)| = |v^\top E(I, u, \cdots, u)|.$

- $|v^\top E(I, I, u, \cdots, u)w| = |E(v, w, u, \cdots, u)|$

*Proof.* It follows

$$|v^\top E(I, u, \cdots, u)| = \left| \sum_{i_1=1}^n v_{i_1} \cdot \left( \sum_{i_2=1}^n \cdots \sum_{i_p=1}^n E_{i_1, i_2, \cdots, i_p} u_{i_2} \cdots u_{i_p} \right) \right|$$

$$= \left| \sum_{i_1=1}^n \sum_{i_2=1}^n \cdots \sum_{i_p=1}^n E_{i_1, i_2, \cdots, i_p} v_{i_1} u_{i_2} \cdots u_{i_p} \right|$$

$$= |E(v, u, \cdots, u)|,$$

where the first step follows from the definition of $E(I, u, \cdots, u)$, the second step follows from the property of summation, and the last step follows from the definition of $E(v, u, \cdots, u)$. $\square$

**Fact B.6.** *If the following conditions hold*

- *$u, v$ are two arbitrary unit vectors.*

- *Suppose $\theta(u, v)$ is in the interval $(0, \pi/2)$.*

*Then $\|u - v\|_2 \le \tan \theta(u, v)$.*

*Proof.* Suppose $\theta(u, v)$ is in the interval $(0, \pi/2)$, so we have

$$\cos \theta(u, v)$$

is in the interval $(0, 1)$.

Let $x = \langle u, v \rangle$.

Therefore, by the definition of $\cos \theta(u, v)$ (see Definition 4.3), we have

$$\cos \theta(u, v) = \langle u, v \rangle$$
$$= x. \tag{8}$$

Accordingly, we have

$$\sin \theta(u, v) = \sqrt{1 - \cos^2 \theta(u, v)}$$
$$= \sqrt{1 - x^2}, \tag{9}$$

where the first step follows from the definition of $\sin \theta(u, v)$ (see Definition 4.3) and the second step follows from Eq. (8). Moreover,

$$\|u - v\|_2^2 = \|u\|_2^2 + \|v\|_2^2 - 2\langle u, v \rangle$$
$$= 1 + 1 - 2x$$
$$= 2 - 2x, \tag{10}$$

where the first step follows from simple algebra, the second step follows from the fact that $u$ and $v$ are unit vectors, and the last step follows from simple algebra. We want to show

$$\|u - v\|_2^2 \le \tan^2 \theta(u, v).$$

It suffices to show

$$2 - 2x \le \tan^2 \theta(u, v)$$

$$= \sin^2 \theta(u,v) / \cos^2 \theta(u,v)$$
$$\leq (1 - x^2)/x^2, \tag{11}$$

where the first step follows from Eq. (10), the second step follows from the definition of $\tan \theta(u,v)$ (see Definition 4.3), and the last step follows from combining Eq. (8) and Eq. (9).

Therefore, it suffices to show

$$(1 - x^2)/x^2 - (2 - 2x) \geq 0$$

when $x \in (0, 1)$.

Let $f : (0, \infty) \to \mathbb{R}$ be defined as

$$f(x) = (1 - x^2)/x^2 - (2 - 2x).$$

Then, the derivative of $f(x)$ is denoted as $f'(x)$, which is as follows

$$f'(x) = \frac{2x^3 - 2}{x^3}.$$

Therefore, when $x = 1$, we have $f'(x) = 0$.

The second derivative of $f$ is

$$f''(x) = \frac{6}{x^4}.$$

Therefore,

$$f''(1) = 6 > 0.$$

Thus, $f(1)$ is a local minimum. In other words, when $x \in (0, 1)$,

$$f(x) = (1 - x^2)/x^2 - (2 - 2x) \geq f(1) = 0,$$

so Eq. (11) is shown to be true.

Thus, we complete the proof. $\square$

### B.3 More Tensor Facts

In this section, we present more tensor properties.

**Fact B.7** (Formal version of Fact 4.4). *If the following conditions hold*

- *Let $p$ be greater than or equal to 3.*
- *Let $A^* = \sum_{j=1}^k \lambda_j v_j^{\otimes p} \in \mathbb{R}^{n^p}$ be an orthogonal tensor.*
- *Let $u \in \mathbb{R}^n$ be a vector.*
- *Let $j \in [k]$.*

*Then, we can get*

$$|v_j^\top A^*(I, u, \cdots, u)| = \lambda_j |v_j^\top u|^{p-1}.$$

*Proof.* For any $j \in [k]$, we have

$$|v_j^\top A^*(I, u, \cdots, u)| = \left| \sum_{i=1}^n v_{j,i} A^*(I, u, \cdots, u)_i \right|$$

$$= \left| \sum_{i=1}^n v_{j,i} \sum_{i_2=1}^n \cdots \sum_{i_p=1}^n A^*_{i,i_2,\cdots,i_p} u_{i_2} \cdots u_{i_p} \right|$$

$$= \left| \sum_{i=1}^{n} v_{j,i} \sum_{i_2=1}^{n} \cdots \sum_{i_p=1}^{n} (\sum_{\ell=1}^{n} \lambda_\ell v_{\ell,i} v_{\ell,i_2} \cdots v_{\ell,i_p}) u_{i_2} \cdots u_{i_p} \right|$$

$$= \left| \sum_{\ell=1}^{k} \lambda_\ell \sum_{i=1}^{n} v_{j,i} v_{\ell,i} \sum_{i_2=1}^{n} \cdots \sum_{i_p=1}^{n} (v_{\ell,i_2} \cdots v_{\ell,i_p}) u_{i_2} \cdots u_{i_p} \right|$$

$$= \left| \lambda_j \sum_{i_2=1}^{n} \cdots \sum_{i_p=1}^{n} (v_{j,i_2} \cdots v_{j,i_p}) u_{i_2} \cdots u_{i_p} \right|$$

$$= \lambda_j |v_j^\top u|^{p-1},$$

where the first step follows from the definition of vector norm, the second step follows from the decomposition of $A^*$ by its definition, the third step follows from the definition of $A^*$, the fourth step follows from reordering the summations, the fifth step follows from taking summations over $\ell$, and the sixth step follows from simple algebra. $\qquad\square$

**Fact B.8** (Formal version of Fact 4.6). *If the following conditions hold*

- *Let $p \geq 3$.*

- *$x, y, u, v \in \mathbb{R}^n$ are four arbitrary unit vectors.*

- *Let $j \in \{0, 1, \cdots, p-2\}$.*

*Then, we can get*

$$\|[x \otimes v^{\otimes(p-1)}](I, u, \cdots, u) - [y \otimes v^{\otimes(p-1)}](I, u, \cdots, u)\|_2 = |\langle u, v \rangle|^{p-1} \cdot \|x - y\|_2 \quad (12)$$

*and*

$$\|[v^{\otimes(1+j)} \otimes x \otimes v^{\otimes(p-2-j)}](I, u, \cdots, u) - [v^{\otimes(1+j)} \otimes y \otimes v^{\otimes(p-2-j)}](I, u, \cdots, u)\|_2$$
$$\leq |\langle u, v \rangle|^{p-2} \cdot \|x - y\|_2. \quad (13)$$

*Proof.* To show Eq. (12), let's analyze the $i$-th entry of the vector

$$[x \otimes v^{\otimes(p-1)}](I, u, \cdots, u) \in \mathbb{R}^n,$$

which can be written as

$$x_i \sum_{i_2=1}^{n} \cdots \sum_{i_p=1}^{n} v_{i_2} \cdots v_{i_p} u_{i_2} \cdots u_{i_p} = x_i \sum_{i_2=1}^{n} v_{i_2} u_{i_2} \cdots \sum_{i_p=1}^{n} v_{i_p} u_{i_p}$$
$$= x_i \langle v, u \rangle^{p-1}, \quad (14)$$

where the first step follows from the property of summation and the second step follows from the definition of the inner product.

In this part, for simplicity, we define

$$\mathrm{LHS} := \|[x \otimes v^{\otimes(p-1)}](I, u, \cdots, u) - [y \otimes v^{\otimes(p-1)}](I, u, \cdots, u)\|_2.$$

By Eq. (14), we have

$$\mathrm{LHS} = \|x_i \langle v, u \rangle^{p-1} - y_i \langle v, u \rangle^{p-1}\|_2.$$

Thus, we get

$$\mathrm{LHS}^2 = \sum_{i=1}^{n} (x_i \langle v, u \rangle^{p-1} - y_i \langle v, u \rangle^{p-1})^2$$

$$= \sum_{i=1}^{n} ((x_i - y_i)(\langle v, u \rangle^{p-1}))^2$$

$$= \sum_{i=1}^{n}(x_i - y_i)^2(\langle v, u\rangle^{p-1})^2$$

$$= \langle v, u\rangle^{2(p-1)}\sum_{i=1}^{n}(x_i - y_i)^2$$

$$= \|x - y\|_2^2 \cdot |\langle v, u\rangle|^{2(p-1)},$$

where the first step follows from the definition of $\|\cdot\|_2$, the second step follows from simple algebra, the third step follows from simple algebra, the fourth step follows from the fact that $i$ is not contained in $\langle v, u\rangle^{2(p-1)}$, and the last step follows from the definition of $\|\cdot\|_2$.

To show Eq. (13), first, we want to show

$$|\langle x - y, u\rangle| \le \|x - y\|_2\|u\|_2$$
$$\le \|x - y\|_2, \tag{15}$$

where the first step follows from the Cauchy–Schwarz inequality and the second step follows from the fact that $u$ is a unit vector so that $\|u\|_2 = 1$.

Then, we analyze the $i$-th entry of the vector

$$[v^{\otimes(1+j)} \otimes x \otimes v^{\otimes(p-2-j)}](I, u, \cdots, u) \in \mathbb{R}^n,$$

which is equivalent to

$$v_i\langle x, u\rangle \cdot \langle v, u\rangle^{p-2}. \tag{16}$$

In this part, we define

$$\text{LHS} := \|[v^{\otimes(1+j)} \otimes x \otimes v^{\otimes(p-2-j)}](I, u, \cdots, u) - [v^{\otimes(1+j)} \otimes y \otimes v^{\otimes(p-2-j)}](I, u, \cdots, u)\|_2.$$

Therefore, based on Eq. (16), we get

$$\text{LHS} = \|v_i\langle x, u\rangle \cdot \langle v, u\rangle^{p-2} - v_i\langle y, u\rangle \cdot \langle v, u\rangle^{p-2}\|_2$$

Thus, we have

$$\text{LHS}^2 = \sum_{i=1}^{n}(v_i\langle x, u\rangle \cdot \langle v, u\rangle^{p-2} - v_i\langle y, u\rangle \cdot \langle v, u\rangle^{p-2})^2$$

$$= \sum_{i=1}^{n}((v_i\langle x, u\rangle - v_i\langle y, u\rangle) \cdot \langle v, u\rangle^{p-2})^2$$

$$= \sum_{i=1}^{n}((v_i\langle x, u\rangle - v_i\langle y, u\rangle)^2 \cdot \langle v, u\rangle^{2(p-2)})$$

$$= \langle v, u\rangle^{2(p-2)} \cdot \sum_{i=1}^{n}(v_i\langle x, u\rangle - v_i\langle y, u\rangle)^2$$

$$= \langle v, u\rangle^{2(p-2)} \cdot \sum_{i=1}^{n}(v_i(\langle x, u\rangle - \langle y, u\rangle))^2$$

$$= \langle v, u\rangle^{2(p-2)} \cdot \sum_{i=1}^{n}(v_i\langle x - y, u\rangle)^2$$

$$= \langle v, u\rangle^{2(p-2)} \cdot \sum_{i=1}^{n}((v_i^2)(\langle x - y, u\rangle^2))$$

$$= \langle x - y, u\rangle^2 \cdot \langle v, u\rangle^{2(p-2)}\sum_{i=1}^{n}(v_i)^2$$

$$= \langle x - y, u\rangle^2 \cdot \langle v, u\rangle^{2(p-2)}$$

$$\leq \|x - y\|_2^2 \cdot \langle v, u \rangle^{2(p-2)},$$

where the first step follows from the definition of $\|\cdot\|_2$, the second step follows from simple algebra, the third step follows from $(ab)^2 = a^2 b^2$, the fourth step follows from the fact that $i$ is not contained in $\langle v, u \rangle^{2(p-2)}$, the fifth step follows from simple algebra, the sixth step follows from the linearity property of the inner product, the seventh step follows from $(ab)^2 = a^2 b^2$, the eighth step follows from the fact that $i$ is not contained in $\langle x - y, u \rangle^2$, the ninth step follows from the fact that $v$ is a unit vector, and the last step follows from Eq. (15). $\qquad\square$

## C    MORE ANALYSIS

In Section C.1, we give the proof to the first part of Theorem 4.9. In Section C.2, we give the proof to the second part of Theorem 4.9. In Section C.3, we give the proof to the third part of Theorem 4.9. In Section C.4, we prove that a few terms are upper-bounded.

### C.1    PART 1 OF THEOREM 4.9

In this section, we present the proof of the first part of Theorem 4.9.

For convenient, we first create some definitions for this section

**Definition C.1.** *We define $B_1 \in \mathbb{R}$ and $B_2 \in \mathbb{R}$ as follows*

$$B_1 := \frac{1}{1 - |v_1^\top \widetilde{E}_{u_t}| / (\lambda_1 |v_1^\top u_t|^{p-1})}$$

*We define*

$$B_2 := \frac{\|V^\top \widetilde{E}_{u_t}\|_2}{\lambda_1 |v_1^\top u_t|^{p-1}}$$

**Lemma C.2** (Part 1 of Theorem 4.9). *If the following conditions hold*

- *Let everything be defined as in Theorem 4.9.*

- *Suppose that all of the assumptions in Theorem 4.9 hold.*

*Then, Eq. (3) hold.*

*Proof.* **Proof of Part 1.**

$V = (v_2, \cdots, v_k, \cdots, v_n) \in \mathbb{R}^{n \times (n-1)}$ is an orthonormal basis and is the complement of $v_1$.

Also, $\widetilde{E}_{u_t} = \widetilde{E}(I, u_t, \cdots, u_t) \in \mathbb{R}^n$.

$\tan \theta(v_1, u_{t+1})$'s upper bound is provided as follows:

$$\tan \theta(v_1, u_{t+1}) = \tan \theta \left( v_1, \frac{A(I, u_t, \cdots, u_t)}{\|A(I, u_t, \cdots, u_t)\|_2} \right)$$
$$= \tan \theta(v_1, A(I, u_t, \cdots, u_t))$$
$$= \tan \theta(v_1, A^*(I, u_t, \cdots, u_t) + \widetilde{E}(I, u_t, \cdots, u_t))$$
$$= \tan \theta(v_1, A^*(I, u_t, \cdots, u_t) + \widetilde{E}_{u_t})$$
$$= \frac{\sin \theta(v_1, A^*(I, u_t, \cdots, u_t) + \widetilde{E}_{u_t})}{\cos \theta(v_1, A^*(I, u_t, \cdots, u_t) + \widetilde{E}_{u_t})}$$
$$= \frac{\|V^\top [A^*(I, u_t, \cdots, u_t) + \widetilde{E}_{u_t}]\|_2}{|v_1^\top [A^*(I, u_t, \cdots, u_t) + \widetilde{E}_{u_t}]|}$$
$$\leq \frac{\|V^\top A^*(I, u_t, \cdots, u_t)\|_2 + \|V^\top \widetilde{E}_{u_t}\|_2}{|v_1^\top A^*(I, u_t, \cdots, u_t)| - |v_1^\top \widetilde{E}_{u_t}|}, \qquad (17)$$

where the first step follows from the definition of $u_{t+1}$, the second step follows from the definition of angle, the third step follows from $A = A^* + \widetilde{E} \in \mathbb{R}^{n^p}$, the fourth step follows from $\widetilde{E}_{u_t} := \widetilde{E}(I, u_t, \cdots, u_t) \in \mathbb{R}^n$, the fifth step follows from the definition of $\tan \theta$, the sixth step follows from $\sin$ and $\cos$, and the seventh step follows from the triangle inequality.

Using Fact 4.7, we can get

$$\|V^\top A^*(I, u_t, \cdots, u_t)\|_2^2 = \sum_{j=2}^k \lambda_j^2 |v_j^\top u_t|^{2(p-1)}$$

$$\leq \left( \max_{j \in [k] \setminus \{1\}} |\lambda_j|^2 |v_j^\top u_t|^{2(p-2)} \right) \cdot \left( \sum_{j=2}^k |v_j^\top u_t|^2 \right), \qquad (18)$$

where the second step follows from $\sum_i a_i b_i \leq (\max_i a_i) \cdot \sum_i b_i$ for all $a, b \in R_{\geq 0}^n$.

Putting it all together, we have

$$\tan \theta(v_1, u_{t+1}) \leq \frac{\|V^\top A^*(I, u_t, \cdots, u_t)\|_2 + \|V^\top \widetilde{E}_{u_t}\|_2}{|v_1^\top A^*(I, u_t, \cdots, u_t)| - |v_1^\top \widetilde{E}_{u_t}|}$$

$$\leq \tan \theta(v_1, u_t) \cdot \frac{(\|V^\top A^*(I, u_t, \cdots, u_t)\|_2 + \|V^\top \widetilde{E}_{u_t}\|_2)/\|V^\top u_t\|_2}{|v_1^\top A^*(I, u_t, \cdots, u_t)|/|v_1^\top u_t| - |v_1^\top \widetilde{E}_{u_t}|/|v_1^\top u_t|}$$

$$\leq \tan \theta(v_1, u_t) \cdot \frac{\max_{j \in [k] \setminus \{1\}} \lambda_j |v_j^\top u_t|^{p-2} + \|V^\top \widetilde{E}_{u_t}\|_2/\|V^\top u_t\|_2}{|v_1^\top A^*(I, u_t, \cdots, u_t)|/|v_1^\top u_t| - |v_1^\top \widetilde{E}_{u_t}|/|v_1^\top u_t|}$$

$$\leq \tan \theta(v_1, u_t) \cdot \frac{\max_{j \in [k] \setminus \{1\}} \lambda_j |v_j^\top u_t|^{p-2} + \|V^\top \widetilde{E}_{u_t}\|_2/\|V^\top u_t\|_2}{\lambda_1 |v_1^\top u_t|^{p-2} - |v_1^\top \widetilde{E}_{u_t}|/|v_1^\top u_t|}$$

$$\leq \tan \theta(v_1, u_t) \cdot \frac{(1/4)\lambda_1 |v_1^\top u_t|^{p-2} + \|V^\top \widetilde{E}_{u_t}\|_2/\|V^\top u_t\|_2}{\lambda_1 |v_1^\top u_t|^{p-2} - |v_1^\top \widetilde{E}_{u_t}|/|v_1^\top u_t|}$$

$$\leq \tan \theta(v_1, u_t) \cdot (1/4) \cdot \underbrace{\frac{1}{1 - |v_1^\top \widetilde{E}_{u_t}|/(\lambda_1 |v_1^\top u_t|^{p-1})}}_{B_1}$$

$$+ \underbrace{\frac{1}{1 - |v_1^\top \widetilde{E}_{u_t}|/(\lambda_1 |v_1^\top u_t|^{p-1})}}_{B_1} \cdot \underbrace{\frac{\|V^\top \widetilde{E}_{u_t}\|_2}{\lambda_1 |v_1^\top u_t|^{p-1}}}_{B_2}$$

$$= \tan \theta(v_1, u_t) \cdot (1/4) \cdot B_1 + B_1 \cdot B_2,$$

where the 1st step comes from Eq. (17), the 2nd step is by $\tan \theta(v_1, u_t) = \frac{\|V^\top u_t\|_2}{|v_1^\top u_t|}$, the 3rd step is because of Equation (18), the 4th step follows from Fact B.7, the 5th step follows from Part 2 of Theorem 4.9, the 6th step follows from simple algebra, and the 7th step follows from the definition of $B_1$ and $B_2$.

We show

**Claim C.3.** *For any $t \in [T]$, we have*

$$|v_1^\top u_0| \leq |v_1^\top u_t|.$$

*Proof.* Based on the assumption from the induction hypothesis, we consider the existence of a sufficiently small constant $c$ being greater than 0 satisfying

$$\tan \theta(v_1, u_t) \leq 0.8 \tan \theta(v_1, u_{t-1}) + c. \qquad (19)$$

Therefore, we can get

$$\tan \theta(v_1, u_t) \leq 0.8 \cdot (0.8 \tan \theta(v_1, u_{t-1}) + c) + c$$

$$\leq 0.8^t \cdot \tan \theta(v_1, u_0) + c \sum_{j=0}^{t-1} 0.8^j$$

$$\leq 0.8^t \cdot \tan \theta(v_1, u_0) + 5c$$

$$\leq \tan \theta(v_1, u_0),$$

where the first step follows from applying Eq. (19) recursively twice, the second step follows from applying Eq. (19) recursively for $t + 1$ times, the third step follows from $\sum_{j=0}^{\infty} 0.8^j \leq 5$, and the last step follows from $\tan \theta(v_1, u_0) = \Omega(1)$.

This result shows

$$\theta(v_1, u_t) \leq \theta(v_1, u_0),$$

so

$$|v_1^\top u_t| = \cos \theta(v_1, u_t) \geq \cos \theta(v_1, u_0) = |v_1^\top u_0|.$$

$\square$

Therefore, $B_1$ and $B_2$ has upper bounds.

**Claim C.4.** $B_1$ *is smaller than or equal to* 1.1.

*Proof.* Let's consider

$$|\widetilde{E}(v_j, u_t, \cdots, u_t)| = |v_j^\top \widetilde{E}(I, u_t, \cdots, u_t)|$$

$$= |v_j^\top \widetilde{E}_{u_t}|.$$

Since

$$|\widetilde{E}(v_j, u_t, \cdots, u_t)| \leq 4\epsilon/\sqrt{n}$$

$$\leq 4c\lambda_1/n^{(p-1)/2}$$

$$\leq 4c\lambda_1 |v_1^\top u_0|^{p-1},$$

where the first step follows from the constraint on $\widetilde{E}$ in Theorem 4.9, the second step follows from $\epsilon \leq c\lambda_1/n^{(p-2)/2}$, and the third step follows from $|v_1^\top u_0| \geq 1/\sqrt{n}$.

Correspondingly, if $c$ can be chosen to be small enough, i.e., $c$ is smaller than $\frac{1}{40}$, using

$$|v_1^\top \widetilde{E}_{u_t}| \leq \lambda_1 |v_1^\top u_0|^{p-1}/10$$

and

$$|v_1^\top u_0| \leq |v_1^\top u_t|,$$

then

$$|v_1^\top \widetilde{E}_{u_t}| \leq \lambda_1 |v_1^\top u_0|^{p-1}/10$$

$$\leq \lambda_1 |v_1^\top u_t|^{p-1}/10. \tag{20}$$

As a result,

$$B_1 \leq \frac{1}{1 - 1/11} = 1.1.$$

$\square$

Next, we bound $B_2$. Let's consider two different cases. The first one is

$$|v_1^\top u_t| \le 1 - \frac{1}{c_0^2 p^2 k^2}$$

and the other is

$$|v_1^\top u_t| > 1 - \frac{1}{c_0^2 p^2 k^2}.$$

If

$$|v_1^\top u_t| \le (1 - \frac{1}{c_0^2 p^2 k^2}),$$

we have

$$B_2 = \frac{\|V^\top \widetilde{E}_{u_t}\|_2}{\lambda_1 |v_1^\top u_t|^{p-1}}$$

$$= \frac{\sqrt{1 - |v_1^\top u_t|^2}}{|v_1^\top u_t|} \cdot \frac{\|V^\top \widetilde{E}_{u_t}\|_2}{\lambda_1 |v_1^\top u_t|^{p-2} \sqrt{1 - |v_1^\top u_t|^2}}$$

$$= \tan\theta(v_1, u_t) \cdot \frac{\|V^\top \widetilde{E}_{u_t}\|_2}{\lambda_1 |v_1^\top u_t|^{p-2} \sqrt{1 - |v_1^\top u_t|^2}}$$

$$\le \tan\theta(v_1, u_t) \cdot \frac{\|\widetilde{E}_{u_t}\|_2}{\lambda_1 |v_1^\top u_t|^{p-2} \sqrt{1 - |v_1^\top u_t|^2}}$$

$$\le \tan\theta(v_1, u_t) \cdot \frac{c_0 p k \|\widetilde{E}_{u_t}\|_2}{\lambda_1 |v_1^\top u_t|^{p-2}},$$

where the first step comes from the definition of $B_2$ (see Definition C.1), the second step follows from splitting the term, the third step follows from $\tan\theta(v_1, u_t) = \frac{\sqrt{1 - |v_1^\top u_t|^2}}{|v_1^\top u_t|}$, the fourth step follows that $\|V^\top \widetilde{E}_{u_t}\|_2 \le \|\widetilde{E}_{u_t}\|_2$, and the last step follows from $1/\sqrt{1 - |v_1^\top u_t|^2} \le c_0 p k$.

We need to bound

$$\frac{c_0 p k \|\widetilde{E}_{u_t}\|_2}{\lambda_1 |v_1^\top u_t|^{p-2}}.$$

Here, we can get

$$\lambda_1 |v_1^\top u_t|^{p-2} \ge \lambda_1 / (n^{(p-2)/2}).$$

On the other hand, utilizing Part 1 of Lemma C.11 and the given assumptions about $\overline{E}$ and $E$, we obtain $\|\widetilde{E}_{u_t}\|_2$

$$c_0 p k \|\widetilde{E}_{u_t}\|_2 \le c_0 p k \cdot 4 p \epsilon.$$

Consequently, whenever we have small enough $\epsilon$ satisfying

$$\epsilon \le \lambda_1 / (n^{(p-2)/2} \cdot 40 \cdot c_0 p^2 k),$$

then

$$B_2 \le 0.1 \tan\theta(v_1, u_t).$$

If

$$|v_1^\top u_t| > 1 - \frac{1}{c_0^2 k^2 p^2},$$

then we have

$$B_2 = \frac{\|V^\top \widetilde{E}_{u_t}\|_2}{\lambda_1 |v_1^\top u_t|^{p-1}}$$

$$\leq \frac{\|V^\top \widetilde{E}_{u_t}\|_2}{\lambda_1 (1 - \frac{1}{c_0^2 p^2 k^2})^{p-1}}$$

$$\leq 3\|V^\top \widetilde{E}_{u_t}\|_2 / \lambda_1$$

$$\leq 3\|\widetilde{E}_{u_t}\|_2 / \lambda_1$$

where the first step follows from the definition of $B_2$, the second step follows from $|v_1^\top u_t| > 1 - \frac{1}{c_0^2 p^2 k^2}$, the third step follows from

$$1/(1 - \frac{1}{c_0^2 p^2 k^2})^{p-1} \leq 3, \forall p \geq 3, k \geq 1, c_0 \geq 1,$$

and the last step follows from $\|V^\top \widetilde{E}_{u_t}\|_2 \leq \|\widetilde{E}_{u_t}\|_2$.

By Part 1 of Corollary C.12, we have $\|\widehat{E}_{u_t}\|_2 \leq 4\epsilon/c_0$. By what we have assumed on $E$ and $\overline{E}$, $\|E_{u_t}\|_2 \leq \epsilon/c_0$ and $\|\overline{E}_{u_t}\|_2 \leq \epsilon/c_0$, which completes the proof of $B_2 \leq 18\epsilon/(c_0\lambda_1)$.

$\square$

## C.2 PART 2 OF THEOREM 4.9

In this section, we present the proof of the second part of Theorem 4.9.

**Lemma C.5** (Part 2 of Theorem 4.9). *If the following conditions hold*

- *Let everything be defined as in Theorem 4.9.*

- *Suppose that all of the assumptions in Theorem 4.9 hold.*

*Then, Eq.* (4) *hold.*

*Proof.* **Proof of Part 2.**

Let $j$ be an arbitrary element in $[k]\backslash\{1\}$.

Then, there exists an lower bound for $\frac{|v_1^\top u_{t+1}|}{|v_j^\top u_{t+1}|}$,

$$\frac{|v_1^\top u_{t+1}|}{|v_j^\top u_{t+1}|} = \frac{|v_1^\top [A^*(I, u_t, \cdots, u_t) + \widetilde{E}_{u_t}]|}{|v_j^\top [A^*(I, u_t, \cdots, u_t) + \widetilde{E}_{u_t}]|}$$

$$\geq \frac{|v_1^\top A^*(I, u_t, \cdots, u_t)| - |v_1^\top \widetilde{E}_{u_t}|}{|v_j^\top A^*(I, u_t, \cdots, u_t)| + |v_j^\top \widetilde{E}_{u_t}|}$$

$$\geq \frac{\lambda_1 |v_1^\top u_t|^{p-1} - |v_1^\top \widetilde{E}_{u_t}|}{\lambda_j |v_j^\top u_t|^{p-1} + |v_j^\top \widetilde{E}_{u_t}|}$$

$$\geq \frac{\lambda_1 |v_1^\top u_t|^{p-1} - \frac{1}{10}\lambda_1 |v_1^\top u_t|^{p-1}}{\lambda_j |v_j^\top u_t|^{p-1} + \frac{1}{10}\lambda_1 |v_1^\top u_t|^{p-1}}$$

$$\geq \frac{\lambda_1 |v_1^\top u_t|^{p-1} - \frac{1}{10}\lambda_1 |v_1^\top u_t|^{p-1}}{\frac{1}{4}\lambda_1 |v_1^\top u_t|^{p-2} |v_j^\top u_t| + \frac{1}{10}\lambda_1 |v_1^\top u_t|^{p-1}}$$

$$= \frac{\frac{9}{10}|v_1^\top u_t|}{\frac{1}{4}|v_j^\top u_t| + \frac{1}{10}|v_1^\top u_t|} \tag{21}$$

where the first step follows from the definition of $u_{t+1}$ (see the statement in Theorem 4.9), the second step follows from the triangle inequality, the third step follows from Fact B.7, the fourth step follows from Eq. (20), the fifth step follows from Part 1 $\lambda_j |v_j^\top u_t|^{p-2} \leq \frac{1}{4}\lambda_1 |v_1^\top u_t|^{p-2}$, and the last step follows from simple algebra.

If

$$|v_j^\top u_t| < |v_1^\top u_t|, \tag{22}$$

then

$$\frac{\lambda_1 |v_1^\top u_{t+1}|^{p-2}}{\lambda_j |v_j^\top u_{t+1}|^{p-2}} = \frac{\lambda_1}{\lambda_j} \left( \frac{|v_1^\top u_{t+1}|}{|v_j^\top u_{t+1}|} \right)^{p-2}$$

$$\geq \frac{\lambda_1}{\lambda_j} \left( \frac{\frac{9}{10}|v_1^\top u_t|}{\frac{1}{4}|v_j^\top u_t| + \frac{1}{10}|v_1^\top u_t|} \right)^{p-2}$$

$$\geq \frac{\lambda_1}{\lambda_j} \left( \frac{\frac{9}{10}|v_j^\top u_t|}{\frac{1}{4}|v_j^\top u_t| + \frac{1}{10}|v_j^\top u_t|} \right)^{p-2}$$

$$= \frac{\lambda_1}{\lambda_j} \left( \frac{\frac{9}{10}}{\frac{1}{4} + \frac{1}{10}} \right)^{p-2}$$

$$\geq \frac{\lambda_1}{\lambda_j} 2^{p-2}, \tag{23}$$

where the first step follows from $\frac{a^x}{b^x} = \left(\frac{a}{b}\right)^x$, the second step follows from Eq. (21), the third step follows from Eq. (22), the fourth step follows from simple algebra, the last step follows from simple algebra.

The final step is a consequence of the fact that $p$ is greater than or equal to $4$. For the case of $p$ being equal to 3, we utilize a better analysis which is similar to the proof of (Wang & Anandkumar, 2016)'s Lemma C.2. Therefore, this approach is applicable for any $p \geq 3$.

If

$$|v_j^\top u_t| \geq |v_1^\top u_t|, \tag{24}$$

then

$$\frac{\lambda_1 |v_1^\top u_{t+1}|^{p-2}}{\lambda_j |v_j^\top u_{t+1}|^{p-2}} \geq \frac{\lambda_1}{\lambda_j} \left( \frac{\frac{9}{10}|v_1^\top u_t|}{\frac{1}{4}|v_j^\top u_t| + \frac{1}{10}|v_1^\top u_t|} \right)^{p-2}$$

$$= \frac{\lambda_1}{\lambda_j} \left( \frac{\frac{9}{10}|v_1^\top u_t||v_j^\top u_t|}{\left(\frac{1}{4}|v_j^\top u_t| + \frac{1}{10}|v_1^\top u_t|\right)|v_1^\top u_t|} \frac{|v_1^\top u_t|}{|v_j^\top u_t|} \right)^{p-2}$$

$$= \frac{\lambda_1}{\lambda_j} \left( \frac{\frac{9}{10}|v_1^\top u_t||v_j^\top u_t|}{\left(\frac{1}{4}|v_j^\top u_t| + \frac{1}{10}|v_1^\top u_t|\right)|v_1^\top u_t|} \right)^{p-2} \left( \frac{|v_1^\top u_t|}{|v_j^\top u_t|} \right)^{p-2}$$

$$= \frac{\lambda_1}{\lambda_j} \left( \frac{\frac{9}{10}|v_j^\top u_t|}{\frac{1}{4}|v_j^\top u_t| + \frac{1}{10}|v_1^\top u_t|} \right)^{p-2} \left( \frac{|v_1^\top u_t|}{|v_j^\top u_t|} \right)^{p-2}$$

$$\geq \frac{\lambda_1}{\lambda_j} \left( \frac{\frac{9}{10}|v_j^\top u_t|}{\frac{1}{4}|v_j^\top u_t| + \frac{1}{10}|v_j^\top u_t|} \right)^{p-2} \left( \frac{|v_1^\top u_t|}{|v_j^\top u_t|} \right)^{p-2}$$

$$\geq \frac{\lambda_1}{\lambda_j} \frac{|v_1^\top u_t|^{p-2}}{|v_j^\top u_t|^{p-2}} \cdot \left( \frac{\frac{9}{10}|v_j^\top u_t|}{\frac{1}{4}|v_j^\top u_t| + \frac{1}{10}|v_j^\top u_t|} \right)^{p-2}$$

$$\geq \frac{\lambda_1}{\lambda_j} \frac{|v_1^\top u_t|^{p-2}}{|v_j^\top u_t|^{p-2}} \cdot 2^{p-2}$$

$$\geq 4 \cdot 2^{p-2},$$

where the first step follows from the second step of Eq. (23), the second step follows from simple algebra, the third step follows from $(ab)^2 = a^2 b^2$, the fourth step follows from simple algebra, the fifth step follows from Eq. (24), the sixth step follows from $\left(\frac{a}{b}\right)^2 = \frac{a^2}{b^2}$, the seventh step follows from the relationship between the third step and the last step of Eq. (23), and the last step follows from Part 1.

$\square$

## C.3 PART 3 OF THEOREM 4.9

In this section, we present the proof of the third part of Theorem 4.9.

**Definition C.6.** *We define $B_3 \in \mathbb{R}$ and $B_4 \in \mathbb{R}$ as follows*

$$B_3 := \frac{1}{1 - |v_1^\top \widetilde{E}_{u_t}|/(\lambda_1 |v_1^\top u_t|^{p-1})}$$

*and*

$$B_4 := \frac{|v_j^\top \widetilde{E}_{u_t}|}{\lambda_1 |v_1^\top u_t|^{p-1}}$$

**Lemma C.7** (Part 3 of Theorem 4.9). *If the following conditions hold:*

- *Let everything be defined as in Theorem 4.9.*

- *Suppose that all of the assumptions in Theorem 4.9 hold.*

*Then, Eq. (5) hold.*

*Proof.* **Proof of Part 3.**

Just like Eq. (21) and Part 2, $\frac{|v_j^\top u_{t+1}|}{|v_1^\top u_{t+1}|}$ can also be upper bounded,

$$
\begin{aligned}
&\frac{|v_j^\top u_{t+1}|}{|v_1^\top u_{t+1}|} \\
&\leq \frac{\lambda_j |v_j^\top u_t|^{p-1} + |v_j^\top \widetilde{E}_{u_t}|}{\lambda_1 |v_1^\top u_t|^{p-1} - |v_1^\top \widetilde{E}_{u_t}|} \\
&= \frac{|v_j^\top u_t|}{|v_1^\top u_t|} \cdot \frac{\lambda_j |v_j^\top u_t|^{p-2} + |v_j^\top \widetilde{E}_{u_t}|/|v_j^\top u_t|}{\lambda_1 |v_1^\top u_t|^{p-2} - |v_1^\top \widetilde{E}_{u_t}|/|v_1^\top u_t|} \\
&= \frac{|v_j^\top u_t|}{|v_1^\top u_t|} \cdot \frac{\lambda_j |v_j^\top u_t|^{p-2}}{\lambda_1 |v_1^\top u_t|^{p-2} - |v_1^\top \widetilde{E}_{u_t}|/|v_1^\top u_t|} + \frac{|v_j^\top \widetilde{E}_{u_t}|}{\lambda_1 |v_1^\top u_t|^{p-1} - |v_1^\top \widetilde{E}_{u_t}|} \\
&= \frac{|v_j^\top u_t|}{|v_1^\top u_t|} \cdot \frac{\lambda_j |v_j^\top u_t|^{p-2}}{\lambda_1 |v_1^\top u_t|^{p-2} - |v_1^\top \widetilde{E}_{u_t}|/|v_1^\top u_t|} + \frac{1}{1 - |v_1^\top \widetilde{E}_{u_t}|/(\lambda_1 |v_1^\top u_t|^{p-1})} \cdot \frac{|v_j^\top \widetilde{E}_{u_t}|}{\lambda_1 |v_1^\top u_t|^{p-1}} \\
&\leq \frac{|v_j^\top u_t|}{|v_1^\top u_t|} \cdot \frac{1}{4} \cdot \frac{\lambda_1 |v_1^\top u_t|^{p-2}}{\lambda_1 |v_1^\top u_t|^{p-2} - |v_1^\top \widetilde{E}_{u_t}|/|v_1^\top u_t|} + \frac{1}{1 - |v_1^\top \widetilde{E}_{u_t}|/(\lambda_1 |v_1^\top u_t|^{p-1})} \cdot \frac{|v_j^\top \widetilde{E}_{u_t}|}{\lambda_1 |v_1^\top u_t|^{p-1}} \\
&= \frac{|v_j^\top u_t|}{|v_1^\top u_t|} \cdot \frac{1}{4} \cdot \underbrace{\frac{1}{1 - |v_1^\top \widetilde{E}_{u_t}|/(\lambda_1 |v_1^\top u_t|^{p-1})}}_{B_3} + \underbrace{\frac{1}{1 - |v_1^\top \widetilde{E}_{u_t}|/(\lambda_1 |v_1^\top u_t|^{p-1})}}_{B_3} \cdot \underbrace{\frac{|v_j^\top \widetilde{E}_{u_t}|}{\lambda_1 |v_1^\top u_t|^{p-1}}}_{B_4},
\end{aligned}
$$

where the first step follows from the relationship between the first step and the third step of Eq. (21), the second step follows from simple algebra, the third step follows from simple algebra, the fourth step follows from simple algebra, and the fifth step follows from Part 1 $\lambda_j |v_j^\top u_t|^{p-2} \leq \frac{1}{4} \lambda_1 |v_1^\top u_t|^{p-2}$, and the last step follows from simple algebra.

Similar to Part 1, we can show $B_3 \leq 1.1$ if

$$|v_1^\top u_t| \geq 1 - \frac{1}{c_0^2 p^2 k^2}.$$

Then, we consider the bound for $B_4$.

There are two different situations, namely,

- Case 1. $|v_1^\top u_t| \le 1 - \frac{1}{c_0^2 p^2 k^2}$

- Case 2. $|v_1^\top u_t| > 1 - \frac{1}{c_0^2 p^2 k^2}$.

If

$$|v_1^\top u_t| \le (1 - \frac{1}{c_0^2 p^2 k^2}),$$

we have

$$
\begin{aligned}
B_4 &= \frac{|v_j^\top \widetilde{E}_{u_t}|}{\lambda_1 |v_1^\top u_t|^{p-1}} \\
&= \frac{\sqrt{1 - |v_1^\top u_t|^2}}{|v_1^\top u_t|} \cdot \frac{|v_j^\top \widetilde{E}_{u_t}|}{\lambda_1 |v_1^\top u_t|^{p-2} \sqrt{1 - |v_1^\top u_t|^2}} \\
&= \tan\theta(v_1, u_t) \cdot \frac{|v_j^\top \widetilde{E}_{u_t}|}{\lambda_1 |v_1^\top u_t|^{p-2} \sqrt{1 - |v_1^\top u_t|^2}} \\
&\le \tan\theta(v_1, u_t) \cdot \frac{c_0 p k |v_j^\top \widetilde{E}_{u_t}|}{\lambda_1 |v_1^\top u_t|^{p-2}},
\end{aligned}
$$

where the 1st step is from the definition of $B_4$, the 2nd step comes from simple algebra, the 3rd step is due to the definition of $\tan\theta(v_1, u_t)$, and the last step follows from $1/\sqrt{1 - |v_1^\top u_t|^2} \le c_0 p k$.

We want to find the bound for

$$\frac{c_0 p k \|\widetilde{E}_{u_t}\|_2}{\lambda_1 |v_1^\top u_t|^{p-2}}.$$

We can get that

$$\lambda_1 |v_1^\top u_t|^{p-2} \ge \lambda_1 / (n^{(p-2)/2}).$$

Additionally, based on Part 2 of Lemma C.11 and what we assumed about $\overline{E}$ and $E$,

$$c_0 p k |v_j^\top \widetilde{E}_{u_t}| \le c_0 p k \cdot 4\epsilon / \sqrt{n}.$$

Therefore, whenever there is a small enough $\epsilon$ satisfying

$$\epsilon \le \lambda_1 \sqrt{n} / (n^{(p-2)/2} \cdot 40 \cdot c_0 p k),$$

then

$$B_4 \le 0.1 \tan\theta(v_1, u_t).$$

If

$$|v_1^\top u_t| > 1 - \frac{1}{c_0^2 k^2 p^2},$$

then we have

$$
\begin{aligned}
B_4 &= \frac{|v_j^\top \widetilde{E}_{u_t}|}{\lambda_1 |v_1^\top u_t|^{p-1}} \\
&\le \frac{|v_j^\top \widetilde{E}_{u_t}|}{\lambda_1 (1 - \frac{1}{c_0^2 p^2 k^2})^{p-1}} \\
&\le 3 |v_j^\top \widetilde{E}_{u_t}| / \lambda_1,
\end{aligned}
$$

where the first step follows from the definition of $B_4$, the second step follows from $|v_1^\top u_t| > 1 - \frac{1}{c_0^2 p^2 k^2}$, the third step follows from $\forall p \ge 3, k \ge 1, c_0 \ge 1$, we have

$$1 / (1 - \frac{1}{c_0^2 p^2 k^2})^{p-1} \le 3.$$

By Part 1 of Corollary C.12, we have

$$|v_j^\top \widehat{E}_{u_t}| \leq 4\epsilon/(c_0\sqrt{n}).$$

Based on what we have assumed about $E$ and $\overline{E}$,

$$|v_j^\top E_{u_t}| \leq \epsilon/(c_0\sqrt{n})$$

and

$$|v_j^\top \overline{E}_{u_t}| \leq \epsilon/(c_0\sqrt{n}).$$

Therefore, the proof of $B_4 \leq 18\epsilon/(c_0\lambda_1\sqrt{n})$ is completed. □

## C.4 $\epsilon$-CLOSE

In this section, we upper bound some terms.

**Definition C.8.** *For any $\epsilon > 0$, we say $\{\widehat{\lambda}_i, \widehat{v}_i\}_{i=1}^k$ is $\epsilon$-close to $\{\lambda_i, v_i\}_{i=1}^k$ if for all $i \in [k]$,*

1. *$|\widehat{\lambda}_i - \lambda_i| \leq \epsilon$.*

2. *$\|\widehat{v}_i - v_i\|_2 \leq \tan\theta(\widehat{v}_i, v_i) \leq \min(\sqrt{2}, \epsilon/(\lambda_i))$.*

3. *$|\widehat{v}_i^\top v_j| \leq \epsilon/(\sqrt{n}\lambda_i), \forall j \in [k]\backslash[i]$.*

**Definition C.9** ($A_i$ and $B_i$)**.** *We define*

$$A_i := \lambda_i a_i^{p-1} v_i - \widehat{\lambda}_i(a_i c_i + \|\widehat{v}_i^\perp\|_2 b_i)^{p-1} c_i$$

*and*

$$B_i := \widehat{\lambda}_i(a_i c_i + \|\widehat{v}_i^\perp\|_2 b_i)^{p-1} \|\widehat{v}_i^\perp\|_2.$$

**Assumption C.10.** *We assume that $\epsilon$ is a real number that satisfies*

$$\epsilon < 10^{-5}\frac{\lambda_k}{p^2 k}.$$

**Lemma C.11.** *If the following conditions hold*

- *For all $i \in [k]$, $\widehat{E}_i = \lambda_i v_i^{\otimes p} - \widehat{\lambda}_i \widehat{v}_i^{\otimes p} \in \mathbb{R}^{n^p}$.*

- *Let $\epsilon > 0$.*

- *$\{\widehat{\lambda}_i, \widehat{v}_i\}_{i=1}^k$ is $\epsilon$-close to $\{\lambda_i, v_i\}_{i=1}^k$.*

- *Let $r \in [k]$.*

- *Let $u \in \mathbb{R}^n$ be an unit vector.*

*Then, we have*

1. *$\left\|\sum_{i=1}^r \widehat{E}_i(I, u, \cdots, u)\right\|_2 \leq 2p\epsilon\kappa^{1/2} + 2\phi\epsilon$.*

2. *For all $[k]\backslash[i]$, $\left|\sum_{i=1}^r \widehat{E}_j(v_j, u, \cdots, u)\right| \leq (2\kappa\epsilon + \phi\epsilon)/\sqrt{n}$.*

*where*

$$\kappa = 2\sum_{i=1}^r |u^\top v_i|^2 \tag{25}$$

*and*

$$\phi = 2k(\epsilon/\lambda_k)^{p-1}. \tag{26}$$

*Proof.* **Proof of Part 1.**

Let $i$ be an arbitrary element in $[r]$.

We have that $\widehat{E}_i$ is the error and it satisfies

$$\widehat{E}_i(I, u, \cdots, u) = \lambda_i (u^\top v_i)^{p-1} v_i - \widehat{\lambda}_i (u^\top \widehat{v}_i)^{p-1} \widehat{v}_i, \tag{27}$$

which is in the span of $\{v_i, \widehat{v}_i\}$.

Also, the span of $\{v_i, \widehat{v}_i\}$ is identical to the span of $\{v_i, \widehat{v}_i^\perp\}$, where

$$\widehat{v}_i^\perp = \widehat{v}_i - (v_i^\top \widehat{v}_i) v_i.$$

is the projection of $\widehat{v}_i$ onto the subspace orthogonal to $v_i$.

Note

$$\|\widehat{v}_i - v\|_2^2 = 2(1 - v_i^\top \widehat{v}_i). \tag{28}$$

For convenient, we define

$$c_i = \langle v_i, \widehat{v}_i \rangle.$$

We can rewrite $c_i$ as follows

$$
\begin{aligned}
c_i &= v_i^\top \widehat{v}_i \\
&= 1 - \|\widehat{v}_i - v_i\|_2^2 / 2 \\
&\geq 0,
\end{aligned}
\tag{29}
$$

the first step follows from definition of $c_i$, the second step follows from Eq. (28), and the last step is because of the assumption that $\|\widehat{v}_i - v_i\|_2 \leq \sqrt{2}$, and it implies that $0 \leq c_i \leq 1$.

We can also get

$$
\begin{aligned}
\|\widehat{v}_i^\perp\|_2^2 &= 1 - c_i^2 \\
&\leq \|\widehat{v}_i - v_i\|_2^2,
\end{aligned}
\tag{30}
$$

which follows from Eq. (29) and the Pythagorean theorem.

For all $p$ being greater than or equal to 3, the following bound can be obtained:

$$
\begin{aligned}
|1 - c_i^p| &= |1 - (1 - \|\widehat{v}_i - v_i\|_2^2 / 2)^p| \\
&\leq \frac{p}{2} \|\widehat{v}_i - v_i\|_2^2,
\end{aligned}
$$

where the 1st step is due to Eq. (29) and the 2nd step is supported by Fact B.1.

We present the definition of $a_i \in \mathbb{R}^n$ and $b_i \in \mathbb{R}^n$:

$$a_i = u^\top v_i$$

and

$$b_i = u^\top (\widehat{v}_i^\perp / \|\widehat{v}_i^\perp\|_2).$$

$\widehat{E}_i(I, u, \cdots, u)$ can be expressed by the coordinate system of $\widehat{v}_i^\perp$ and $v_i$:

$$
\begin{aligned}
&\widehat{E}_i(I, u, \cdots, u) \\
={}& \lambda_i (u^\top v_i)^{p-1} v_i - \widehat{\lambda}_i (u^\top \widehat{v}_i)^{p-1} \widehat{v}_i \\
={}& \lambda_i a_i^{p-1} v_i - \widehat{\lambda}_i (a_i c_i + \|\widehat{v}_i^\perp b_i\|_2)^{p-1} (c_i v_i + \widehat{v}_i^\perp) \\
={}& \underbrace{(\lambda_i a_i^{p-1} v_i - \widehat{\lambda}_i (a_i c_i + \|\widehat{v}_i^\perp\|_2 b_i)^{p-1} c_i)}_{A_i} \cdot v_i - \underbrace{\widehat{\lambda}_i (a_i c_i + \|\widehat{v}_i^\perp\|_2 b_i)^{p-1} \|\widehat{v}_i^\perp\|_2}_{B_i} \cdot \widehat{v}_i^\perp / \|\widehat{v}_i^\perp\|_2 \\
={}& A_i \cdot v_i - B_i \cdot (\widehat{v}_i^\perp / \|\widehat{v}_i^\perp\|_2),
\end{aligned}
\tag{31}
$$

where the first step follows from Eq. (27), the second step follows from the definition of $a_i$ and $b_i$, the third step follows from simple algebra, and the last step follows from the definition of $A_i$ and $B_i$ (see Definition C.9).

We can express the overall error by:

$$\left\|\sum_{i=1}^{r} \widehat{E}_i(I, u, \cdots, u)\right\|_2^2 = \left\|\sum_{i=1}^{r} A_i v_i - \sum_{i=1}^{t} B_i(\widehat{v}_i^{\perp}/\|\widehat{v}_i^{\perp}\|_2)\right\|_2^2$$

$$\leq 2\left\|\sum_{i=1}^{r} A_i v_i\right\|_2^2 + 2\left\|\sum_{i=1}^{r} B_i(\widehat{v}_i^{\perp}/\|\widehat{v}_i^{\perp}\|_2)\right\|_2^2$$

$$\leq 2\sum_{i=1}^{r} A_i^2 + 2\left(\sum_{i=1}^{r} |B_i|\right)^2, \tag{32}$$

where the first step follows from Eq. (31), the second step follows from triangle inequality, and the third step comes from the definition of the $\ell_2$ norm.

We have

$$\|\widehat{v}_i^{\perp}\|_2^2 \leq \|\widehat{v}_i - v_i\|_2^2$$
$$\leq \epsilon/\lambda_i, \tag{33}$$

where the first step follows from Eq. (30) and the second step follows from Definition C.8.

By using

$$|\lambda_i - \widehat{\lambda}_i| \leq \epsilon, \tag{34}$$

we first try to bound $A_i$ for $|b_i|$ being smaller than 1 and $c_i \in [0, 1]$,

$$|A_i| = |\lambda_i a_i^{p-1} - \widehat{\lambda}_i(a_i c_i + \|\widehat{v}_i^{\perp}\|_2 b_i)^{p-1} c_i|$$

$$\leq |\lambda_i a_i^{p-1} - \widehat{\lambda}_i c_i^p a_i^{p-1}| + \sum_{j=1}^{p-1} \widehat{\lambda}_i c_i \binom{(p-1)}{j} |a_i c_i|^{(p-1)-j} \|\widehat{v}_i^{\perp}\|_2^j$$

$$= |\lambda_i a_i^{p-1} - \widehat{\lambda}_i c_i^p a_i^{p-1} + \widehat{\lambda}_i a_i^{p-1} - \widehat{\lambda}_i a_i^{p-1}| + \sum_{j=1}^{p-1} \widehat{\lambda}_i c_i \binom{(p-1)}{j} |a_i c_i|^{(p-1)-j} \|\widehat{v}_i^{\perp}\|_2^j$$

$$\leq |\lambda_i a_i^{p-1} - \widehat{\lambda}_i a_i^{p-1}| + |\widehat{\lambda}_i a_i^{p-1} - \widehat{\lambda}_i c_i^p a_i^{p-1}| + \sum_{j=1}^{p-1} \widehat{\lambda}_i c_i \binom{(p-1)}{j} |a_i c_i|^{(p-1)-j} \|\widehat{v}_i^{\perp}\|_2^j$$

$$\leq |\lambda_i a_i^{p-1} - \widehat{\lambda}_i a_i^{p-1}| + |(1 - c_i^p)\widehat{\lambda}_i a_i^{p-1}| + \sum_{j=1}^{p-1} \widehat{\lambda}_i c_i \binom{(p-1)}{j} |a_i c_i|^{(p-1)-j} \|\widehat{v}_i^{\perp}\|_2^j$$

$$\leq |\lambda_i a_i^{p-1} - \widehat{\lambda}_i a_i^{p-1}| + |(1 - c_i^p)\widehat{\lambda}_i a_i^{p-1}| + \sum_{j=1}^{p-1} \widehat{\lambda}_i c_i \binom{(p-1)}{j} |a_i c_i|^{(p-1)-j} (\epsilon/\lambda_i)^j$$

$$\leq \epsilon |a_i|^{p-1} + \frac{p}{2}(\epsilon/\lambda_i)^2 \widehat{\lambda}_i |a_i|^{p-1} + \sum_{j=1}^{p-1} \widehat{\lambda}_i \binom{(p-1)}{j} |a_i|^{(p-1)-j} (\epsilon/\lambda_i)^j, \tag{35}$$

where the first step follows from the definition of $A_i$ (see Definition C.9), the second step follows from binomial theorem and $|b_i| < 1$, the third step follows from adding and subtracting the same thing, the fourth step follows from triangle inequality, the fifth step follows from simple algebra, the sixth step follows from Eq. (33), and the last step follows from Eq. (34).

Note that the second term of Eq. (35) can be bounded as

$$\frac{p}{2}(\epsilon/\lambda_i)^2 \widehat{\lambda}_i |a_i|^{p-1} \leq \frac{p}{2}(\epsilon/\lambda_i)^2 (|\widehat{\lambda}_i - \lambda_i| + |\lambda_i|) |a_i|^{p-1}$$

$$= \frac{p}{2}(\epsilon/\lambda_i)^2 |\widehat{\lambda}_i - \lambda_i||a_i|^{p-1} + \frac{p}{2}(\epsilon/\lambda_i)^2 |\lambda_i||a_i|^{p-1}$$

$$\leq \frac{p}{2}(\epsilon/\lambda_i)^2 \epsilon |a_i|^{p-1} + \frac{p}{2}(\epsilon/\lambda_i)^2 |\lambda_i||a_i|^{p-1}$$

$$\leq \frac{p}{2}(\epsilon/\lambda_i)^2 \epsilon |a_i|^{p-1} + \frac{p}{2}(\epsilon/\lambda_i) \cdot \epsilon |a_i|^{p-1}$$

$$\leq \frac{1}{100k}\epsilon |a_i|^{p-1}, \tag{36}$$

where the first step follows from triangle inequality, the second step follows from simple algebra, the third step follows from Eq. (34), the fourth step follows from Eq. (33), and the last step follows from Assumption C.10.

We can separate the third term of Eq. (35) into two components

1. $j \in \{1, \cdots, (p-1)/2\}$ and

2. $j \in \{(p-1)/2, \cdots, (p-1)\}$.

Consider the first component:

$$\sum_{j=1}^{(p-1)/2} \widehat{\lambda}_i \binom{(p-1)}{j} |a_i|^{(p-1)-j} (\epsilon/\lambda_i)^j$$

$$= \widehat{\lambda}_i(p-1)|a_i|^{p-2}\epsilon/\lambda_i + \sum_{j=2}^{(p-1)/2} \widehat{\lambda}_i(p-1)^j |a_i|^{(p-1)/2}(\epsilon/\lambda_i)^j$$

$$\leq 2(p-1)\epsilon|a_i|^{p-2} + \sum_{j=2}^{(p-1)/2} \widehat{\lambda}_i(p-1)^j |a_i|^{(p-1)/2}(\epsilon/\lambda_i)^j$$

$$= 2(p-1)\epsilon|a_i|^{p-2} + \widehat{\lambda}_i|a_i|^{(p-1)/2} \sum_{j=2}^{(p-1)/2} (p-1)^j (\epsilon/\lambda_i)^j$$

$$\leq 2(p-1)\epsilon|a_i|^{p-2} + \widehat{\lambda}_i|a_i|^{(p-1)/2} \sum_{j=0}^{\infty} \left(\frac{1}{2}\right)^j$$

$$\leq 2(p-1)\epsilon|a_i|^{p-2} + \widehat{\lambda}_i\epsilon\frac{\epsilon(p-1)^2}{\lambda_i^2} \cdot 2 \cdot |a_i|^{(p-1)/2}$$

$$\leq 2(p-1)\epsilon|a_i|^{p-2} + \frac{1}{100k}\epsilon|a_i|^{(p-1)/2},$$

where the first step is by expanding the summation term, the second step is because of the fact that $\widehat{\lambda}_i \leq 2\lambda_i$, the third step is supported by $\sum_i ca_i = c\sum_i a_i$, the fourth step follows from the fact that each term of $\sum_{j=2}^{(p-1)/2}(p-1)^j(\epsilon/\lambda_i)^j$ is bounded by the corresponding term of $\sum_{j=0}^{\infty}\left(\frac{1}{2}\right)^j$, the fifth step follows from Fact B.2, the sixth step follows from Assumption C.10 and $\widehat{\lambda}_i \leq 2\lambda_i$.

Similarly, we can bound the second component,

$$\sum_{j=(p-1)}^{(p-1)/2} \widehat{\lambda}_i \binom{(p-1)}{j} |a_i|^{(p-1)-j} (\epsilon/\lambda_i)^j$$

$$= \widehat{\lambda}_i \cdot 1 \cdot 1 \cdot (\epsilon/\lambda_i)^{p-1} + \sum_{j=p}^{(p-1)/2} \widehat{\lambda}_i \binom{(p-1)}{j} |a_i|^{(p-1)-j} (\epsilon/\lambda_i)^j$$

$$= 2\epsilon^{p-1}/\lambda_i^{p-2} + \sum_{j=p}^{(p-1)/2} \widehat{\lambda}_i \binom{(p-1)}{j} |a_i|^{(p-1)-j} (\epsilon/\lambda_i)^j$$

$$\leq 2\epsilon^{p-1}/\lambda_i^{p-2} + \binom{(p-1)}{(p-1)/2} \sum_{j=p}^{(p-1)/2} \widehat{\lambda}_i |a_i|^{(p-1)-j} (\epsilon/\lambda_i)^j$$

$$\leq 2\epsilon^{p-1}/\lambda_i^{p-2} + \binom{(p-1)}{(p-1)/2} \widehat{\lambda}_i |a_i|^{(p-1)/2} \sum_{j=p}^{(p-1)/2} (\epsilon/\lambda_i)^j$$

$$\leq \frac{1}{100k}\epsilon + \binom{(p-1)}{(p-1)/2} \widehat{\lambda}_i |a_i|^{(p-1)/2} \sum_{j=p}^{(p-1)/2} (\epsilon/\lambda_i)^j$$

$$\leq \frac{1}{100k}\epsilon + \binom{(p-1)}{(p-1)/2} \widehat{\lambda}_i |a_i|^{(p-1)/2} (\epsilon/\lambda_i)^{(p-1)/2} \sum_{j=p}^{(p-1)/2} (\epsilon/\lambda_i)^{j-(p-1)/2}$$

$$\leq \frac{1}{100k}\epsilon + 2\binom{(p-1)}{(p-1)/2} \widehat{\lambda}_i |a_i|^{(p-1)/2} (\epsilon/\lambda_i)^{(p-1)/2}$$

$$\leq \frac{1}{100k}\epsilon + 4\binom{(p-1)}{(p-1)/2} \lambda_i |a_i|^{(p-1)/2} (\epsilon/\lambda_i)^{(p-1)/2}$$

$$\leq \frac{1}{100k}\epsilon + \frac{1}{100k}\epsilon|a_i|$$

where the first step comes from expanding the summation term, the second step can be gotten from $\widehat{\lambda}_i \leq 2\lambda_i$, the third step can be supported by $\max_j\{\binom{(p-1)}{j}\} = \binom{(p-1)}{(p-1)/2}$, the fourth step follows from $\sum_i ca_i = c\sum_i a_i$, the fifth step follows from Assumption C.10, the sixth step follows from simple algebra, the seventh step follows from the Fact B.3, the eighth step follows from $\widehat{\lambda}_i \leq 2\lambda_i$, and the last step follows from Assumption C.10.

Thus, putting it all together, we get

$$|A_i| \leq \epsilon|a_i| + \frac{1}{10k}\epsilon|a_i| + (p-1)\epsilon|a_i| + \frac{1}{100k}\epsilon$$

which implies that

$$|A_i|^2 \leq 2\left((\epsilon|a_i|)^2 + (\frac{1}{10k}\epsilon|a_i|)^2 + ((p-1)\epsilon|a_i|)^2 + (\frac{1}{100k}\epsilon)^2\right) \tag{37}$$

Next, we need to find the bound of $B_i$,

$$|B_i| = |\widehat{\lambda}_i\|\widehat{v}_i^\perp\|_2 (a_i c_i + \|\widehat{v}_i^\perp\|_2 b_i)^{p-1}|$$

$$\leq \widehat{\lambda}_i\|\widehat{v}_i^\perp\|_2 \sum_{j=0}^{p-1} \binom{(p-1)}{j} |a_i c_i|^{p-1-j} \|\widehat{v}_i^\perp\|_2^j$$

$$\leq \widehat{\lambda}_i(\epsilon/\lambda_i) \sum_{j=0}^{p-1} \binom{(p-1)}{j} |a_i|^{p-1-j} (\epsilon/\lambda_i)^j$$

$$= \widehat{\lambda}_i(\epsilon/\lambda_i)|a_i|^{p-1} + \widehat{\lambda}_i(\epsilon/\lambda_i) \sum_{j=1}^{p-1} \binom{(p-1)}{j} |a_i|^{p-1-j} (\epsilon/\lambda_i)^j, \tag{38}$$

where the first step follows from the definition of $B_i$ (see Definition C.9), the second step follows from binomial theorem and $|b_i| < 1$, the third step follows from $\|\widehat{v}_i^\perp\|_2 \leq \epsilon/\lambda_i$, and the last step follows from extracting the first term from the summation.

Note that the first term of Eq. (38) is

$$\widehat{\lambda}_i(\epsilon/\lambda_i)|a_i|^{p-1},$$

which can be bounded as

$$\widehat{\lambda}_i(\epsilon/\lambda_i)|a_i|^{p-1} \leq (\epsilon + \lambda_i)(\epsilon/\lambda_i)|a_i|^{p-1}$$

$$= \epsilon |a_i|^{p-1} + (\epsilon^2/\lambda_i)|a_i|^{p-1}$$

$$\leq \epsilon |a_i|^{p-1} + \frac{1}{100k}\epsilon |a_i|^{p-1}$$

where the first step follows from simple algebra, and the second step follows from Eq. (36).

The second term of Eq. (38) is

$$\widehat{\lambda}_i(\epsilon/\lambda_i) \sum_{j=1}^{p-1} \binom{(p-1)}{j} |a_i|^{p-1-j}(\epsilon/\lambda_i)^j,$$

We can separate this into two components

1. $j \in \{1, \cdots, (p-1)/2\}$ and

2. $j \in \{(p-1)/2, \cdots, (p-1)\}$.

The first component is:

$$(\epsilon/\lambda_i) \sum_{j=1}^{(p-1)/2} \widehat{\lambda}_i \binom{(p-1)}{j} |a_i|^{(p-1)-j}(\epsilon/\lambda_i)^j$$

$$= (\epsilon/\lambda_i)\widehat{\lambda}_i(p-1)|a_i|^{p-2}\epsilon/\lambda_i + (\epsilon/\lambda_i) \sum_{j=2}^{(p-1)/2} \widehat{\lambda}_i(p-1)^j|a_i|^{(p-1)/2}(\epsilon/\lambda_i)^j$$

$$\leq \frac{1}{100k}\epsilon |a_i|^{p-2} + \frac{1}{100k}\epsilon |a_i|^{(p-1)/2}$$

where the first step comes from expanding the summation term and the second step is by Assumption C.10.

The second component is:

$$(\epsilon/\lambda_i) \sum_{j=(p-1)}^{(p-1)/2} \widehat{\lambda}_i \binom{(p-1)}{j} |a_i|^{(p-1)-j}(\epsilon/\lambda_i)^j$$

$$= (\epsilon/\lambda_i)\widehat{\lambda}_i \cdot 1 \cdot 1 \cdot (\epsilon/\lambda_i)^{p-1} + (\epsilon/\lambda_i) \sum_{j=p}^{(p-1)/2} \widehat{\lambda}_i \binom{(p-1)}{j} |a_i|^{(p-1)-j}(\epsilon/\lambda_i)^j$$

$$\leq \frac{\phi}{k}\epsilon + \frac{1}{100k}\epsilon |a_i|$$

where the first step follows from expanding the summation term, and the second step follows from $\phi = 2k(\epsilon/\lambda_k)^{p-1}$ and Assumption C.10.

Putting it all together, we have

$$|B_i| \leq \epsilon |a_i|^2 + \frac{1}{100k^2}\epsilon |a_i| + \frac{\phi}{k}\epsilon$$

Taking the summation over all the $r$ terms on both sides, we obtain

$$\sum_{i=1}^{r} |B_i| \leq \epsilon\kappa + \frac{1}{100k^2}\epsilon \sum_{i=1}^{t} |a_i| + \phi\epsilon.$$

Using

$$(x + y + z)^2 \leq 3(x^2 + y^2 + z^2),$$

we have

$$(\sum_{i=1}^{r}|B_i|)^2 \leq 3\left((\epsilon\kappa)^2 + (\frac{1}{100k}\epsilon\sum_{i=1}^{t}|a_i|)^2 + (\phi\epsilon)^2\right)$$

$$\leq 3\left((\epsilon\kappa)^2 + (\frac{1}{100k}\epsilon)^2\kappa k + (\phi\epsilon)^2\right), \tag{39}$$

where the last step follows from $(\sum_{i=1}^{t}|a_i|)^2 \leq \kappa k$.

Recall that

$$\kappa = \sum_{i=1}^{t}|a_i|^2 \leq 1.$$

In general, we can get

$$\left\|\sum_{i=1}^{r}\widehat{E}_i(I, u, \cdots, u)\right\|_2^2 = 2\sum_{i=1}^{r}|A_i|^2 + 2(\sum_{i=1}^{r}|B_i|)^2$$

$$\leq 4\left(\epsilon^2 \cdot \kappa + (\frac{1}{10k}\epsilon)^2\kappa + (p-1)^2\epsilon^2\kappa + (\frac{1}{100\sqrt{k}}\epsilon)^2\right)$$

$$+ 2(\sum_{i=1}^{r}|B_i|)^2$$

$$\leq 4\left(\epsilon^2 \cdot \kappa + (\frac{1}{10k}\epsilon)^2\kappa + (p-1)^2\epsilon^2\kappa + (\frac{1}{100\sqrt{k}}\epsilon)^2\right)$$

$$+ 4\left((\epsilon\kappa)^2 + (\frac{1}{100k}\epsilon)^2\kappa k + (\phi\epsilon)^2\right)$$

$$\leq 4p^2\epsilon^2\kappa + 4\phi^2\epsilon^2$$

where the first step follows from Eq. (32), the second step follows from Eq. (37), the third step follows from Eq. (39), and the last step follows from simple algebra.

The desired bound is given by this equation.

**Proof of Part 2.**

Let $j$ be an arbitrary element of $[k]\backslash[i]$.

Now, we can get

$$\left|\sum_{i=1}^{r}\widehat{E}_i(v_j, u, \cdots, u)\right| \leq \sum_{i=1}^{r}|\widehat{E}_i(v_j, u, \cdots, u)|$$

$$= \sum_{i=1}^{r}\widehat{\lambda}_i|\widehat{v}_i^\top u|^{p-1}|\widehat{v}_i^\top v_j|$$

$$\leq \sum_{i=1}^{t}\widehat{\lambda}_i|\widehat{v}_i^\top u|^{p-1}\frac{\epsilon}{\sqrt{n}\lambda_i},$$

for the 1st step, we use the triangle inequality, for the 2nd step, we utilize the fact that $\langle v_j, v_i \rangle = 0, \forall i \neq j$, and for the last step, we employ the third part of the definition of $\epsilon$-close.

Now, we analyze the bound for $\sum_{i=1}^{r}\frac{\widehat{\lambda}_i}{\lambda_i}|\widehat{v}_i^\top u|^{p-1}$:

$$\sum_{i=1}^{r}\frac{\widehat{\lambda}_i}{\lambda_i}|\widehat{v}_i^\top u|^{p-1} \leq \sum_{i=1}^{r}\frac{\widehat{\lambda}_i}{\lambda_i}(|v_i^\top u| + |(v_i - \widehat{v}_i)^\top u|)^{p-1}$$

$$\leq \sum_{i=1}^{r}\frac{\widehat{\lambda}_i}{\lambda_i}(|v_i^\top u| + \|v_i - \widehat{v}_i\|_2)^{p-1}$$

$$\leq \sum_{i=1}^{r} (1 + \epsilon/\lambda_i) \cdot (|v_i^\top u| + \epsilon/\lambda_i)^{p-1}$$

$$\leq 2 \sum_{i=1}^{r} (|v_i^\top u| + \epsilon/\lambda_i)^{p-1}$$

$$\leq 2 \sum_{i=1}^{r} 2^{p-2} \cdot |v_i^\top u|^{p-1} + 2^{p-2} \cdot (\epsilon/\lambda_i)^{p-1}$$

$$\leq 2 \sum_{i=1}^{r} 2|v_i^\top u|^2 + (2\epsilon/\lambda_i)^{p-1}$$

$$\leq 2 \sum_{i=1}^{r} 2|v_i^\top u|^2 + 2k(2\epsilon/\lambda_k)^{p-1}$$

$$= 2\kappa + \phi.$$

where the first step follows from the triangle inequality, the second step follows from Cauchy-Scharwz inequality and $\|u\|_2 \leq 1$, the third step follows from Eq. (33), the fourth step follows from $\epsilon/\lambda_i \leq 2$, the fifth step follows from Fact B.1, the sixth step follows from $|v_i^\top u| < 1/4$ and $p \geq 3$, the seventh step follows from $\lambda_k \leq \lambda_i$, and the last step follows from the definition of $\phi$ and $\kappa$ (see Eq. (26) and Eq. (25)).

Then, we complete the proof with the desired bound $(2\kappa + \phi)\epsilon/\sqrt{n}$.

$\square$

**Corollary C.12.** *If the following conditions hold:*

- *For all $i \in [k]$, let $\widehat{E}_i = \lambda_i v_i^{\otimes p} - \widehat{\lambda}_i \widehat{v}_i^{\otimes p} \in \mathbb{R}^{n^p}$.*

- *Let $c_0 \geq 1$.*

- *Let $r \in [k]$.*

- *Let $\epsilon \leq \lambda_k/(2c_0 k)$.*

- *Suppose that $\{\widehat{\lambda}_i, \widehat{v}_i\}_{i=1}^{k}$ is $\epsilon$-close to $\{\lambda_i, v_i\}_{i=1}^{k}$.*

- *$u \in \mathbb{R}^n$ is an unit vector.*

- *Suppose $|u^\top v_{r+1}| \geq 1 - \frac{1}{c_0^2 p^2 k}$.*

- *Let $\kappa = \sum_{i=1}^{r} |u^\top v_i|^2$.*

- *Let $\phi = 2k(\epsilon/\lambda_k)^{p-1}$.*

*Then,*

*1.* $\left\| \sum_{i=1}^{r} \widehat{E}_i(I, u, \cdots, u) \right\|_2 \leq 2p\epsilon\kappa^{1/2} + 2\phi\epsilon \leq 4\epsilon/c_0.$

*2.* $\forall j \in [k]\backslash[i], \left| \sum_{i=1}^{r} \widehat{E}_j(v_j, u, \cdots, u) \right| \leq (2\kappa\epsilon + \phi\epsilon)/\sqrt{n} \leq 4\epsilon/(c_0\sqrt{n}).$

*Proof.* Based on Fact B.4, we can get that for any arbitrary $i$ in $[r]$,

$$\kappa = \sum_{i=1}^{r} |u^\top v_i|^2$$

$$\leq k \cdot 1/(c_0^2 p^2 k)$$

$$= 1/c_0^2 p^2,$$

where the first step follows from definition of $\kappa$, the second step follows from $r \leq k$ and $\max_i |u^\top v_i|^2 \leq 1/(c_0^2 p^2 k)$, and the last step follows from simple algebra.

This implies

$$2p\epsilon\sqrt{\kappa} \leq 2/c_0.$$

We can also bound $\phi$,

$$
\begin{aligned}
\phi &= 2k(\epsilon/\lambda_k)^{p-1} \\
&\leq 2k(1/2c_0 k)^{p-1} \\
&\leq 2k(1/(2c_0 k))^2 \\
&\leq 1/c_0.
\end{aligned}
$$

where the 1st step can be gotten from the definition of $\phi$, the 2nd step is because of $\epsilon \leq \lambda_k/(2c_0 k)$ (from Corollary statement), the 3rd step is due to $p \geq 2$, and the 4th step can be seen from $2k \geq 1$.

Therefore, we complete our proof. $\qquad\square$

## D COMBINE

In this section, we present Theorem D.1 and Theorem D.2 and prove them.

**Theorem D.1** (Arbitrary order robust tensor power method, formal version of Lemma 3.1). *If the following conditions hold*

- *Let $p$ be greater than or equal to $3$.*

- *Let $k$ be greater than or equal to $1$.*

- *Let $\lambda_i > 0$.*

- *With $n \geq k$, $\{v_1, \ldots, v_k\} \subseteq \mathbb{R}^n$ is an orthonormal basis vectors.*

- *Let $A = A^* + E \in \mathbb{R}^{n^p}$ be an arbitrary tensor satisfying $A^* = \sum_{i=1}^k \lambda_i v_i^{\otimes p}$.*

- *Suppose that $\lambda_1$ is the greatest values in $\{\lambda_i\}_{i=1}^k$.*

- *Suppose that $\lambda_k$ is the smallest values in $\{\lambda_i\}_{i=1}^k$.*

- *The outputs obtained from the robust tensor power method are $\{\widehat{\lambda}_i, \widehat{v}_i\}_{i=1}^k$.*

- *$E$ satisfies that $\|E\| \leq \epsilon/(c_0\sqrt{n})$.*

- *$T = \Omega(\log(\lambda_1 n/\epsilon))$.*

- *$L = \Omega(k \log(k))$.*

- *$c_0 \geq 100$ and $c > 0$*

- *For all $\epsilon$ satisfying $\epsilon \in (0, c\lambda_k/(c_0 p^2 k n^{(p-2)/2})$.*

*Then, with probability at least $9/10$, there exists a permutation $\pi : [k] \to [k]$, such that $\forall i \in [k]$,*

$$|\lambda_i - \widehat{\lambda}_{\pi(i)}| \leq \epsilon, \quad \|v_i - \widehat{v}_{\pi(i)}\|_2 \leq \epsilon/\lambda_i. \tag{40}$$

*Proof.* Let $E \in \mathbb{R}^{n^p}$ be the original noise.

Let

$$\widehat{E}_i = \lambda_i v_i^{\otimes p} - \widehat{\lambda}_i \widehat{v}_i^{\otimes p} \in \mathbb{R}^{n^p}$$

be the deflation noise.

$\overline{E} \in \mathbb{R}^{n^p}$ represents the sketch noise.

$\widetilde{E}$ represents the "true" noise, including all the original, deflation and sketch noises.

As a result, for the $t + 1$ step, we analyze $A^* + \widetilde{E}$, which is a tensor satisfying

$$\widetilde{E} = E + \sum_{i=1}^{t} \widehat{E}_i + \overline{E}.$$

There is no need for us to consider $\overline{E}$, the sketch noise. However, to prove a stronger statement, we do not regard $\overline{E}$ to be equal to 0, but only assume that it is bounded, namely

$$\|\overline{E}\| \leq \epsilon/(c_0 \sqrt{n}). \tag{41}$$

We use mathematical induction to proof this.

**Base case.**

Let $i = 1$.

For the 1st step, we have that $\widehat{\lambda}_1 \in \mathbb{R}$ and $\widehat{v}_1 \in \mathbb{R}^n$.

As Part 2 of Definition C.8, we show

$$\|\widehat{v}_1 - v_1\|_2$$

is bounded.

Then, as Part 1 of Definition C.8, we show

$$|\widehat{\lambda}_1 - \lambda_1|$$

is bounded.

At the end, as Part 3 of Definition C.8, we show

$$|\widehat{v}_i^\top v_j|$$

is bounded.

**Bounding $|\widehat{v}_1 - v_1|$.**

We have

$$\begin{aligned}
\tan \theta(u_0, v_1) &= \sin \theta(u_0, v_1)/\cos \theta(u_0, v_1) \\
&= \sqrt{1 - \langle u_0, v_1 \rangle^2}/\langle u_0, v_1 \rangle \\
&= \sqrt{\frac{1 - \langle u_0, v_1 \rangle^2}{\langle u_0, v_1 \rangle^2}} \\
&= \sqrt{\frac{1}{\langle u_0, v_1 \rangle^2} - 1} \\
&\leq \sqrt{\frac{1}{\langle u_0, v_1 \rangle^2}} \\
&= \frac{1}{\langle u_0, v_1 \rangle} \\
&\leq \sqrt{n}, \tag{42}
\end{aligned}$$

where the first step follows from Definition 4.3, the second step follows from Definition 4.3, the third step follows from simple algebra, the fourth step follows from simple algebra, the fifth step follows from simple algebra, the sixth step follows from simple algebra, and the last step follows from Lemma 4.8.

$t^*$ represents the condition for

$$|u_{t^*}^\top v_1| = 1 - \frac{1}{c_0^2 p^2 k^2}. \tag{43}$$

We know

$$\begin{aligned}
\|u_{t^*} - v_1\|_2^2 &= \|u_{t^*}\|_2^2 + \|v_1\|_2^2 - 2\langle u_{t^*}, v_1 \rangle \\
&= 1 + 1 - 2\langle u_{t^*}, v_1 \rangle \\
&= 2 - 2|u_{t^*}^\top v_1| \\
&= 2 - 2(1 - \frac{1}{c_0^2 p^2 k^2}) \\
&= 2/(c_0^2 p^2 k^2),
\end{aligned}$$

where the first step follows from simple algebra, the second step follows from the fact that $u_{t^*}$ and $v_1$ are unit vectors, the third step is because the inner product is positive, the fourth step follows from Eq. (43), and the last step follows from simple algebra.

We can upper bound

$$\begin{aligned}
\|u_{t^*} - v_1\|_2 &\le \tan\theta(u_{t^*}, v_1) \\
&\le 0.8 \tan\theta(u_{t^*-1}, v_1) \\
&\le \cdots \\
&\le 0.8^{t^*} \tan\theta(u_0, v_1) \\
&\le 0.8^{t^*} \sqrt{n},
\end{aligned}$$

where the first step is due to Fact B.6, the second can be seen from Part 1 of Theorem 4.9, the second last step can be gotten from Part 1 of Theorem 4.9, and the last step follows from Eq. (42).

After that, we let

$$t^* = \Omega(\log(nkpc_0)) = \Omega(\log(c_0 n)).$$

For $\|u_T - v_1\|_2$, we can show

$$\begin{aligned}
\|u_T - v_1\|_2 &\le 0.8 \tan\theta(u_T, v_1) + 18\epsilon/(c_0\lambda_1) \\
&\le \cdots \\
&\le 0.8^{T-t^*} \tan(u_{t^*}, v_1) + 5 \cdot 18\epsilon/(c_0\lambda_1),
\end{aligned}$$

where the first step follows from Part 1 of Theorem 4.9, and the last step follows from recursively applying Part 1 of Theorem 4.9.

To guarantee

$$\|u_T - v_1\|_2 \le \epsilon/\lambda_1,$$

we let

$$T - t^* = \Omega(n\lambda_1/\epsilon)$$

and $c_0 \ge 100$.

Therefore, we achieve the intended property as outlined in Part 2 of Definition C.8.

**Bounding $|\widehat{\lambda}_1 - \lambda_1|$.**

It remains to bound $|\widehat{\lambda}_1 - \lambda_1|$.

$$\begin{aligned}
|\widehat{\lambda}_1 - \lambda_1| &= |[A^* + \widetilde{E}](\widehat{v}_1, \cdots, \widehat{v}_1) - \lambda_1| \\
&\le |\widetilde{E}(\widehat{v}_1, \cdots, \widehat{v}_1)| + |A^*(\widehat{v}_1, \cdots, \widehat{v}_1) - \lambda_1| \\
&= |\widetilde{E}(\widehat{v}_1, \cdots, \widehat{v}_1)| + \left|\left[\sum_{i=1}^k \lambda_i v_i^{\otimes p}\right](\widehat{v}_1, \cdots, \widehat{v}_1) - \lambda_1\right|
\end{aligned}$$

$$\leq \underbrace{|\widetilde{E}(\widehat{v}_1, \cdots, \widehat{v}_1)|}_{B_5} + \underbrace{|\lambda_1|v_1^\top \widehat{v}_1|^p - \lambda_1|}_{B_6} + \underbrace{\sum_{j=2}^{k} \lambda_j |v_j^\top \widehat{v}_1|^p}_{B_7}, \tag{44}$$

where the first step follows from the definition of $\widehat{\lambda}_1$, the second step follows from the triangle inequality, the third step follows from

$$A^* = \sum_{i=1}^{k} \lambda_i v_i^{\otimes p},$$

and the last step follows from the triangle inequality.

For the term $B_5$, we have

$$\begin{aligned} B_5 &= |\widetilde{E}(\widehat{v}_1, \cdots, \widehat{v}_1)| \\ &\leq |E(\widehat{v}_1, \cdots, \widehat{v}_1)| + |\overline{E}(\widehat{v}_1, \cdots, \widehat{v}_1)| \\ &\leq \|E\| + |\overline{E}(\widehat{v}_1, \cdots, \widehat{v}_1)| \\ &\leq \epsilon/(c_0\sqrt{n}) + \epsilon/(c_0\sqrt{n}) \\ &\leq \epsilon/12, \end{aligned} \tag{45}$$

where the first step follows from the definition of $B_5$ (see Eq. (44)), the second step follows from triangle inequality, the third step follows from the definition of tensor spectral norm, the fourth step follows from Eq. (41), and the last step follows from $c_0 \geq 100$ and $n$ is greater than or equal to 1.

We still need to find the upper bound of $B_6$ and $B_7$.

$$\begin{aligned} B_6 &= |\lambda_1 \cdot |v_1^\top \widehat{v}_1|^p - \lambda_1| \\ &= \lambda_1 - \lambda_1(1 - \frac{1}{2}\|v_1 - \widehat{v}_1\|_2^2)^p \\ &\leq \lambda_1 p \frac{1}{2}\|v_1 - \widehat{v}_1\|_2^2 \\ &\leq p\epsilon^2/(2\lambda_1) \\ &\leq \epsilon/12, \end{aligned} \tag{46}$$

where the first step follows from the definition of $B_6$ (see Eq. (44)), the second step follows from $v_1^\top \widehat{v}_1 = 1 - \frac{1}{2}\|v_1 - \widehat{v}_1\|_2^2$, the third step comes from $\|v_1 - \widehat{v}_1\|_2^2 \ll 1$, the fourth step is because of $\|v_1 - \widehat{v}_1\|_2 \leq \epsilon/\lambda_1$, and the last step follows from $p\epsilon/(2\lambda_1) \leq 1/12$.

For $B_7$, we have

$$\begin{aligned} B_7 &= \sum_{j=2}^{k} \lambda_j |v_j^\top \widehat{v}_1|^p \\ &\leq \sum_{j=2}^{k} \lambda_j (\epsilon/(\sqrt{n}\lambda_j))^p \\ &= \epsilon \sum_{j=2}^{k} (\epsilon/(\lambda_j\sqrt{n}))^{p-1} \\ &\leq \epsilon/4, \end{aligned} \tag{47}$$

where the first step follows from the definition of $B_7$ (see Eq. (44)), the second step follows from Part 3 of Definition C.8, the third step follows from simple algebra, and the last step is due to $(\epsilon/\lambda_k)^{p-1} \leq 1/(4k)$.

Let

$$\epsilon < \frac{1}{4}k^{1/(p-1)}\lambda_k.$$

Finally, combining everything together, we can get

$$|\widehat{\lambda}_1 - \lambda_1| \leq B_5 + B_6 + B_7$$
$$\leq \epsilon/12 + \epsilon/12 + \epsilon/4$$
$$\leq \epsilon,$$

where the first step follows from Eq. (44), the second step follows from combining Eq. (45), Eq. (46), and Eq. (47), and the last step follows from simple algebra.

**Bounding $|\widehat{v}_1^\top v_j|$.**

Let $j$ be an arbitrary element in $\{2, \cdots, k\}$.

Let $t^*$ be the least integer satisfying

$$|v_1^\top u_{t^*}| \geq 1 - \frac{1}{c_0^2 p^2 k^2},$$

which implies

$$|v_j^\top u_{t^*}| \leq \frac{1}{c_0 p k}.$$

By Part 3 of Theorem 4.9, we have

$$|v_j^\top u_{t^*}|/|v_1^\top u_{t^*}| \leq 0.8|v_j^\top u_{t^*-1}|/|v_1^\top u_{t^*-1}|$$
$$\leq \cdots$$
$$\leq 0.8^{t^*} \cdot |v_j^\top u_0|/|v_1^\top u_0|$$
$$\leq 0.8^{t^*} \cdot |v_j^\top u_0|/(1/\sqrt{n})$$
$$\leq 0.8^{t^*} \cdot 1/(1/\sqrt{n}),$$

where the third step follows from recursively applying Part 3 of Theorem 4.9, the fourth step follows from Lemma 4.8 and the last step follows from the fact that $|v_j^\top u_0|$ is at most 1.

Let

$$t^* = \Omega(\log c_0 n).$$

When $T > t^*$, we have

$$|v_j^\top u_T|/|v_1^\top u_T| \leq 0.8^{T-t^*}|v_j^\top u_{t^*}|/|v_1^\top u_{t^*}| + 5 \cdot 18\epsilon/(c_0 \lambda_1 \sqrt{n}).$$

Let

$$T = \Omega(\log(n\lambda_1/\epsilon))$$

and $c_0 \geq 100$ to ensure

$$|v_j^\top u_T| \leq \epsilon/(\lambda_1 \sqrt{n}).$$

**Inductive case.**

Let $i = r + 1$.

Suppose the first $r$ cases holds.

To show the $r + 1$ case also hold, we first consider the "true" noise, which is

$$\widetilde{E} = E + \sum_{i=1}^{r} E_i + \overline{E} \in \mathbb{R}^{n^p}.$$

We explain how to bound

$$\|\widehat{v}_{r+1} - v_{r+1}\|_2,$$

(for Definition C.8, Part 2).

Then, we show how to bound

$$|\widehat{\lambda}_{r+1} - \lambda_{r+1}|$$

as Part 1 of Definition C.8.

In the end, we show how to bound

$$|v_{r+1}^\top v_j|$$

as Part 3 of Definition C.8.

**Bounding** $\|\widehat{v}_{r+1} - v_{r+1}\|_2$**.**

Except for letting

$$T = \Omega(\log(n\lambda_{t+1}/\epsilon)),$$

other parts of the proof are the same as the ones in the base case.

**Bounding** $|\widehat{\lambda}_{r+1} - \lambda_{r+1}|$**.**

Let $A^*$ and $\widetilde{E}$ be

$$A^* = \sum_{i=t+1}^k \lambda_i v_i^{\otimes p}$$

and

$$\widetilde{E} = E + \overline{E} + \sum_{i=1}^t \widehat{E}_i.$$

Therefore, we have

$$|\widehat{\lambda}_{t+1} - \lambda_{t+1}|$$

satisfying

$$
\begin{aligned}
|\widehat{\lambda}_{r+1} - \lambda_{r+1}| &= |[A^* + \widetilde{E}](\widehat{v}_{r+1}, \cdots, \widehat{v}_{r+1}) - \lambda_{r+1}| \\
&\leq |\widetilde{E}(\widehat{v}_{r+1}, \cdots, \widehat{v}_{r+1})| + |A^*(\widehat{v}_{r+1}, \cdots, \widehat{v}_{r+1}) - \lambda_{r+1}| \\
&= |\widetilde{E}(\widehat{v}_{r+1}, \cdots, \widehat{v}_{r+1})| + \left| \left[ \sum_{i=r+1}^k \lambda_i v_i^{\otimes p} \right] (\widehat{v}_{r+1}, \cdots, \widehat{v}_{r+1}) - \lambda_{r+1} \right| \\
&\leq \underbrace{|\widetilde{E}(\widehat{v}_{r+1}, \cdots, \widehat{v}_{r+1})|}_{B_8} + \underbrace{|\lambda_{r+1}|v_{r+1}^\top \widehat{v}_{r+1}|^p - \lambda_{r+1}|}_{B_9} + \underbrace{\sum_{j=r+2}^k \lambda_j |v_j^\top \widehat{v}_{r+1}|^p}_{B_{10}}.
\end{aligned}
$$

where the first step follows from the definition of $\widehat{\lambda}_{r+1}$, the second step follows from triangle inequality, the third step follows from $A^* = \sum_{i=r+1}^k \lambda_i v_i^{\otimes p}$, and the last step follows from the triangle inequality.

We need to analyze $B_8$,

$$
\begin{aligned}
B_8 &= |\widetilde{E}(\widehat{v}_{r+1}, \cdots, \widehat{v}_{r+1})| \\
&= |E(\widehat{v}_{r+1}, \cdots, \widehat{v}_{r+1})| + |\overline{E}(\widehat{v}_{r+1}, \cdots, \widehat{v}_{r+1})| + |\sum_{i=1}^r \widehat{E}_i(\widehat{v}_{r+1}, \cdots, \widehat{v}_{r+1})| \\
&\leq \epsilon/(c_0\sqrt{n}) + \epsilon/(c_0\sqrt{n}) + 4\epsilon/(c_0\sqrt{n}) \\
&\leq \epsilon/12,
\end{aligned}
$$

where the first step follows from the definition of $B_8$, the second step follows from the triangle inequality, the third step follows from Eq. (41), the last step follows from $c_0 \geq 100, n \geq 1$.

$B_9$ and $B_{10}$ can be bounded in a similar way as the base case.

**Bounding $|\widehat{v}_{r+1}^\top v_j|$.**

Let $j$ be an arbitrary element in $\{r+2, \cdots, k\}$. Then, the proof is the same as the base case. $\qquad \square$

**Theorem D.2** (Fast Tensor Power Method via Sketching, formal version of Theorem 1.1)**.** *If the following conditions hold*

- *Let $A = A^* + E \in \mathbb{R}^{n^p}$ be an arbitrary tensor satisfying $A^* = \sum_{i=1}^k \lambda_i v_i^{\otimes p}$.*

- *Suppose that $\lambda_1$ is the greatest values in $\{\lambda_i\}_{i=1}^k$.*

- *Suppose that $\lambda_k$ is the smallest values in $\{\lambda_i\}_{i=1}^k$.*

- *The outputs obtained from the robust tensor power method are $\{\widehat{\lambda}_i, \widehat{v}_i\}_{i=1}^k$.*

- *$E$ satisfies that $\|E\| \leq \epsilon/(c_0\sqrt{n})$.*

- *$T = \Omega(\log(\lambda_1 n/\epsilon))$.*

- *$L = \Omega(k \log(k))$.*

- *$c_0 \geq 100$ and $c > 0$*

- *For all $\epsilon$ satisfying $\epsilon \in (0, c\lambda_k/(c_0 p^2 k n^{(p-2)/2})$.*

*Then, our algorithm uses $\widetilde{O}(n^p)$ spaces, runs in $O(TL)$ iteration, and in each iteration it takes $\widetilde{O}(n^{p-1})$ time and then with probability at least $1 - \delta$, there exists a permutation*

$$\pi : [k] \to [k],$$

*such that $\forall i \in [k]$,*

$$|\lambda_i - \widehat{\lambda}_{\pi(i)}| \leq \epsilon, \quad \|v_i - \widehat{v}_{\pi(i)}\|_2 \leq \epsilon/\lambda_i.$$

*Proof.* It follows by combining Theorem D.1 and Lemma 4.2. $\qquad \square$

## LLM USAGE DISCLOSURE

LLMs were used only to polish language, such as grammar and wording. These models did not contribute to idea creation or writing, and the authors take full responsibility for this paper's content.

