# OpenReview forum: "Tensor Power Methods: Faster and Robust for Arbitrary Order"
_ICLR.cc/2026/Conference — Submitted to ICLR 2026_

### Official Review · Reviewer_NGvr · 2025-10-31

**Soundness:** 3
**Presentation:** 1
**Contribution:** 2
**Rating:** 4
**Confidence:** 3

**Summary:**

This paper presents a theoretical study on tensor power methods for finding eigenvectors and eigenvalues of tensors of order p \ge 3. The authors propose an algorithm (Algorithm 1) and provide rigorous proofs for the approximation errors. The paper aims to generalize existing power method analyses, which have been primarily limited to third-order tensors (e.g., Wang & Anandkumar, 2016).

**Strengths:**

The main strength of this work lies in its theoretical completeness. The proposed results extend the scope of tensor power methods beyond the well-studied third-order case, filling a notable gap in the literature. The proofs are detailed, and the technical rigor contributes to the theoretical understanding of tensor eigenvalue computations.

**Weaknesses:**

However, I have several concerns regarding the clarity and significance of the contribution. First, the paper is difficult to follow in several parts due to unclear definitions, missing explanations of notations, and imprecise mathematical exposition. I will discuss these issues in detail in the “Questions” section.

Second, while the contribution is well stated, I am not fully convinced of its overall importance for the following reasons.

(1) The theoretical results rely on a strong assumption that A^* must be constructed using orthonormal basis vectors (see the fourth condition in Theorem D.1, which is also used in Theorem 1.1). This assumption, although standard in CPD analyses for third-order tensors, seems overly restrictive for higher-order tensors due to the over-determinacy of tensor eigenvectors—there are typically many more eigenvectors than the ambient dimension. I could be convinced if the authors provided numerical evidence showing that this assumption is not too strong in practice, or additional theoretical arguments addressing this concern.

(2) The complexity analysis in Theorem 1.1 shows that the computational cost of the proposed algorithm grows exponentially with the tensor order p. This raises doubts about the claim of having a “fast” algorithm, especially since the main focus is on tensors of order greater than three.

(3) I could not clearly identify what new techniques were introduced to handle the higher-order case compared to prior works. The proofs appear to rely on more algebraic derivations rather than fundamentally new ideas. If the authors can clearly highlight the non-trivial extensions and explain how the proof differs from the third-order case, this concern could be easily addressed.

**Questions:**

1. In line 44, regarding the citation of CP decomposition, why not cite the original work by Frank (1927)? Is there a specific reason for omitting it?
2. For Theorem 1.1, could you elaborate on how the proposed algorithm achieves computational efficiency compared to existing methods?
3. In lines 119–122, the phrase “and this combination is the only… the rank-1 tensor that… is not possible” is unclear. Does this refer to the uniqueness property of CP decomposition?
4. In line 178, the notation v_j is not clearly explained. What does the subscript j represent?
5. In line 197, what does “part 1” refer to, and in line 200, what does V on the right-hand side of the inequality denote? These should be clarified in the main text, not only in the appendix, to improve readability.
6. In the algorithm section, what is the relationship between A and A_i for i \in [n]?
7.  In Lemma 4.2, the definition of \delta should also be presented in the main paper for completeness.
8. In line 996, what is meant by an “orthogonal tensor”? Please provide a formal definition.


⸻

LLM Usage Disclosure:
Parts of this review were refined using a large language model (OpenAI GPT-5) to improve grammar and readability. All substantive assessments, critiques, and questions reflect my own independent judgment.

---

> ### Author Response · Authors · 2025-11-20
> **First response to reviewer NGvr, part 1/2**
>
> ### Your review was both uplifting and insightful. Thank you for your kind assessment!
> We have addressed all your concerns below. We also made corresponding revisions in our latest pdf. All changes from the originally submitted version are highlighted in blue in the revised PDF.
> ### Weakness 1 & Question 2, 3
>
> Thank you for your question regarding the empirical validation of our algorithms. While empirical validation would be interesting, demonstrating these guarantees experimentally would require substantial engineering effort and domain‑specific datasets that lie outside the paper’s scope, since our primary contribution is intentionally theoretical. We focused on rigorous analysis and matching upper/lower‑bound proofs.
>
> We would like to clarify that our work is purely a theoretical study in the tensor power method, which has a well-established place in top venues. Our paper is aligned with theoretical studies such as [1, 2, 3, 4], focusing on the theoretical aspects of tensor power method and optimization.
>
> ### Weakness 2
>
> Thank you for raising this point. Our notion of “fast” is relative to existing tensor power methods for fixed order p, which is the standard regime in tensor decomposition: in almost all applications p is a small constant (typically $p=3$ or 4), determined by the input data rather than tuned by the algorithm. To our knowledge, our work is the fastest tensor power method for arbitrary p. We hope our work could inspire future algorithms.
>
> ### Weakness 3
>
> We sincerely appreciate your continued engagement and thoughtful critiques. We would like to clarify the novelty and contributions of our work respectfully.
>
> #### Generalizing from $p \leq 3$ to arbitrary $p$.
> We agree that the properties of eigenvectors are generalized from [4], from bounded order ($p \leq 3$) to general order $p$, the proof requires a much different analysis and sketching technique.
>
> Our analysis derives p-order bounds on $\tan \theta(v_1, u_{t+1})$, and on the $\frac{\lambda_1|v_1^\top u_{t+1}|^{p-2}}{\lambda_j|v_j^\top u_{t+1}|^{p-2}} \ge 4\cdot 2^{p-2}$, and a more comprehensive analysis of the term $B_4 := \frac{|v_j^\top \widetilde{E}_{u_t}|}{\lambda_1|v_1^\top u_t|^{p-1}}$.We upper bound higher-order new properties that hold for all $p \geq 3$, which is not covered by the original proof in [4].
>
> #### Sketching techniques for faster algorithm
> Meanwhile, we provide a faster algorithm via non-trivial sketching techniques for the problem of tensor decomposition in [5], which achieves time complexity $\widetilde{O}(\epsilon^{-2}n^{p-1} + n^2k)$, compared to the original time of $O(nkdLR)$,where $R = \Omega(\log d + \log \log(\lambda_{\max}/\epsilon)), L = \Omega(\max\{K_0,k\}\log(\max\{K_0,k\}))$.
>
> ### Question 1
>
> Thank you for this remark. In the revision we cite the "The expression of a tensor or a polyadic as a sum of products" by Frank L. Hitchcock as the earliest reference for CP decomposition.
>
> ### Question 2
>
> Thank you for raising this question. The computational efficiency mainly contributed by sketching techniques. For technical details, please refer to section 3 and section 4.
>
> ### Question 3
>
> Thank you for pointing out this ambiguity. Yes, the sentence is intended to refer to the uniqueness property of CP decomposition: under standard identifiability conditions [6], a tensor’s representation as a sum of rank-1 tensors is unique up to permutation and rescaling of the components.
>
> ### Question 4
>
> Thank you for raising this question. Subscript j represent an arbitrary index of the orthonormal eigenvectors of the original, unperturbed tensor $A^*$.
>
> ### Question 5
>
> Thanks for raising this question. “Part 1” refers to Part 1 of Theorem 4.9. And the definition of $V$ is $V = (v_2, \cdots,v_k,\cdots, v_n) \in \mathbb{R}^{n\times (n-1)}$ is an orthonormal basis and is the complement of $v_1$. In the revision, we relabel 1, 2, and 3 as Part 1, Part 2, and Part 3 in Theorem 4.9 for clarity, and we will also clarify the definition of $V$ in the main text when discussing the recoverability analysis.

---

> ### Author Response · Authors · 2025-11-20
> **First response to reviewer NGvr, part 2/2**
>
> ### Question 6
>
> We clarify that $A_i$ represents the $i$-th slice of the unperturbed signal tensor A^* along its first mode (where $A = A^* + E$).
>
> And the definition of $A^\*$ is $A^* = \sum_{j=1}^k \lambda_j v_j^{\otimes p}$.
>
> By fixing the first index to $i$, $A_i$ becomes a tensor of order $(p-1)$ with the decomposition $A_i = \sum_{j=1}^k \alpha_{i,j} v_j^{\otimes (p-1)}$
>
> ### Question 7
>
> Thanks for your careful observation. You are absolutely right. The $\epsilon, \delta \in (0,1/2)$ are parameters we accidentally forget to mention. In Lemma 4.2 there exists a randomized data structure with the operations with the probability of success at least $1 - \delta$. In the revised version we add this definition in the main paper for completeness. Thanks again for pointing out this question.
>
> ### Question 8
>
> Thank you for pointing out this ambiguity. A tensor $A^* \in \mathbb{R}^{d \times d \times d}$ is said to have a CP (Candecmp/Parafac) rank $k$ if it can be (minimally) written as the sum of $k$ rank-1 tensors:
>
> \begin{equation}
>     A^* = \sum_{i \in [k]} \lambda_i \, v_i \otimes v_i \otimes v_i,
>     \lambda_i \in \mathbb{R}^+, v_i \in \mathbb{R}^d.
> \end{equation}
>
> A tensor is said to be orthogonally decomposable if in the above decomposition $\langle v_i, v_j \rangle = 0$ for $i \neq j$. In the revision, we will add this definition to our paper when the term is first introduced.
>
> ### Reference
>
> [1] "Guarantees for Alternating Least Squares in Overparameterized Tensor Decomposition", Dionysis Arvanitakis, Vaidehi Srinivas, Aravindan Vijayaraghavan, NeurIPS 2025
>
> [2] "Lower and Upper Bounds on the Pseudo-Dimension of Tensor Network Models
> ", Behnoush Khavari, Guillaume Rabusseau, NeurIPS 2021
>
> [3] "Beyond Lazy Training for Over-parameterized Tensor Decomposition", Xiang Wang, Chenwei Wu, Jason D. Lee, Tengyu Ma, Rong Ge, NeurIPS 2020
>
> [4] "Fast and Guaranteed Tensor Decomposition via Sketching", Yining Wang, Hsiao-Yu Tung, Alexander Smola, Anima Anandkumar, NeurIPS 2015
>
> [5] "Online and Differentially-Private Tensor Decomposition", Yining Wang, Animashree Anandkumar, NeurIPS 2016
>
> [6] "Foundations of the parafac procedure: Models and conditions for an ‘explanatory’ multimodal factor analysis", Richard A. Harshman, UCLA Working Papers in Phonetics, 1970

---

> > ### Comment · Reviewer_NGvr · 2025-11-27
> >
> > Thank you for your response to my concern.
> >
> > Regarding your reply to W1, my intention is not to request purely numerical experiments to validate your assumption. What I mean is that I expect to see stronger evidence (I suggested the numerical expiermnts because it seems simpler) demonstrating that the orthogonality assumption is reasonable in practice. I understand that this assumption is widely used in theoretical analyses of CP decomposition, and I also know it has a long history of simplifying proofs. That might have been acceptable a decade ago when most works focused on order-3 tensors. However, you empheisesd in the paper your work targets at high order tensors!! It's much diferent. For practical tensors with even modest order, this assumption feels too far from reality.
> >
> > I also appreciate your comment that _“our work is purely a theoretical study in the tensor power method, which has a well-established place in top venues.”_ However, I do not believe that “top venues” should be treated as a universal justification in every situation. Research should aim to expand the boundaries of these venues, not use them as a shield for defense.

---

> > > ### Comment · Reviewer_NGvr · 2025-11-27
> > >
> > > Thanks for addressing my concern about the term “fast.” I’m fine with any naming choice; my only point is to highlight the exponential complexity of the methods with respect to the tensor order.
> > >
> > > I also appreciate your revisions to the previously unclear points. Since clarity was one of my main concerns, these improvements are good.

---

### Official Review · Reviewer_ZQ8e · 2025-10-31

**Soundness:** 3
**Presentation:** 2
**Contribution:** 2
**Rating:** 4
**Confidence:** 4

**Summary:**

This paper presents a generalization of the robust Tensor Power Method (TPM) for Canonical Polyadic (CP) tensor decomposition to arbitrary tensor orders $p \geq 3$. Previous work (e.g., Anandkumar et al., 2014; Wang & Anandkumar, 2016) provided robust guarantees and convergence proofs for third-order tensors under bounded noise. The authors extend these results by developing a theoretical framework for higher-order tensors, establishing formal convergence and error bounds, and proposing a fast sketch-based implementation using randomized Hadamard transforms (RHT). The algorithm achieves a claimed runtime of $\tilde{O}(n^{p-1})$ per iteration and maintains robustness guarantees under noise. The paper provides detailed proofs and lemmas supporting the generalization.

**Strengths:**

**Originality:**

The paper addresses a formal generalization of the robust TPM from third-order tensors to arbitrary-order tensors, filling a theoretical gap in tensor decomposition literature.

While the idea itself is incremental, the generalization requires careful handling of tensor algebra, noise bounds, and convergence proofs, which the authors provide comprehensively.

**Quality:**

The mathematical derivations are technically sound and complete, with proofs and lemmas that appear consistent and rigorous.

The authors clearly define all assumptions, norms, and quantities, ensuring reproducibility of the theoretical work.

**Clarity:**

Key results (Lemma 3.1, Theorem 4.9) are stated cleanly, and the appendices provide formal proofs.

Despite its technical density, the paper is generally self-contained.

**Significance:**

The generalization to arbitrary-order tensors contributes to the theoretical completeness of tensor power methods.

The work may serve as a reference for theoreticians studying high-order tensor decompositions, even if it has limited immediate practical impact.

**Weaknesses:**

**Limited Novelty Beyond Extension**

The core algorithmic idea follows directly from prior robust TPM frameworks (Anandkumar et al., 2014; Wang & Anandkumar, 2016).

The generalization to arbitrary order relies mainly on extending existing proofs by induction, without introducing fundamentally new insights or techniques.

Sketching via randomized Hadamard transforms is also well-established and applied here in a standard way.

**No Empirical Validation**

The paper makes claims of being “faster” and “robust,” but presents no experiments or synthetic evaluations to substantiate these claims.

Runtime scaling (Õ(n^{p-1})) and robustness under realistic noise conditions remain theoretical and unverified.

**Exposition is overly technical and lacks insight.**

**Questions:**

1) Can you include small synthetic experiments (for example p = 3,4,5) showing actual runtime scaling or robustness under controlled noise? This would strengthen the “faster” and “robust” claims. More empirical experiments (synthetic or real) would also be a good addition to the paper.

2) For what practical range of tensor order p and dimension n does your algorithm remain computationally feasible?

3) How sensitive is convergence to initialization? Does the generalization change the probability of successful recovery compared to p=3?

4) Could this framework handle non-orthogonally decomposable or overcomplete tensors? Is that theoretically tractable within your current proof structure?

---

> ### Author Response · Authors · 2025-11-20
> **First response to reviewer ZQ8e**
>
> ### Thank you for highlighting the strengths of our work. Your perspective is deeply valued.
> ### Weakness 2 & Question 1,2
>
> Thank you for your question regarding the empirical validation of our algorithms. While empirical validation would be interesting, demonstrating these guarantees experimentally would require substantial engineering effort and domain‑specific datasets that lie outside the paper’s scope, since our primary contribution is intentionally theoretical. We focused on rigorous analysis and matching upper/lower‑bound proofs.
>
> We would like to clarify that our work is purely a theoretical study in the tensor power method, which has a well-established place in top venues. Our paper is aligned with theoretical studies such as [1, 2, 3, 4], focusing on the theoretical aspects of tensor power method and optimization.
>
> ### Weakness 1
>
> We sincerely appreciate your continued engagement and thoughtful critiques. We would like to clarify the novelty and contributions of our work respectfully.
>
> #### Generalizing from $p \leq 3$ to arbitrary $p$.
> We agree that the properties of eigenvectors are generalized from [4], from bounded order ($p \leq 3$) to general order $p$, the proof requires a much different analysis and sketching technique.
>
> Our analysis derives p-order bounds on $\tan \theta(v_1, u_{t+1})$, and on the $\frac{\lambda_1|v_1^\top u_{t+1}|^{p-2}}{\lambda_j|v_j^\top u_{t+1}|^{p-2}} \ge 4\cdot 2^{p-2}$, and a more comprehensive analysis of the term $B_4 := \frac{|v_j^\top \widetilde{E}_{u_t}|}{\lambda_1|v_1^\top u_t|^{p-1}}$. We upper bound higher-order new properties that hold for all $p \geq 3$, which are not covered by the original proof in [4].
>
> #### Sketching techniques for faster algorithm
> Meanwhile, we provide a faster algorithm via non-trivial sketching techniques for the problem of tensor decomposition in [4], which achieves time complexity $\widetilde{O}(\epsilon^{-2}n^{p-1} + n^2k)$, compared to the original time of $O(nkdLR)$, where $R = \Omega(\log d + \log \log(\lambda_{\max}/\epsilon)), L = \Omega(\max\{K_0,k\}\log(\max\{K_0,k\}))$.
>
> ### Question 3
>
> Thank you for raising these questions. Our work is purely theoretical, the sensitivity to initialization is out of our scope.
>
> Regarding the probability of successful recovery, Lemma 4.8 shows that for any $\eta \in (0, 1/2)$ we can choose parameters so that the algorithm succeeds with probability at least $1 - \eta$. This is exactly the same as Lemma C.1 in [5] where $p \leq 3$.
>
> ### Question 4
> Thank you for raising this question. Our current theory is developed in the orthogonally decomposable setting. In the undercomplete case where the components $\{v\}^k_{i=1}$ are linearly independent, one can apply the invertible whitening transform [6] to obtain an orthogonal tensor.
> The overcomplete tensors are not covered by our present proof structure: in that case, whitening is no longer invertible, and one cannot reduce to an orthogonal model in a straightforward way. Extending our framework to overcomplete tensors would require additional efforts beyond the current analysis, and we view this as an interesting direction for future work. Thanks again for pointing out this critical question.
>
> Reference
>
> [1] "Guarantees for Alternating Least Squares in Overparameterized Tensor Decomposition", Dionysis Arvanitakis, Vaidehi Srinivas, Aravindan Vijayaraghavan, NeurIPS 2025
>
> [2] "Lower and Upper Bounds on the Pseudo-Dimension of Tensor Network Models
> ", Behnoush Khavari, Guillaume Rabusseau, NeurIPS 2021
>
> [3] "Beyond Lazy Training for Over-parameterized Tensor Decomposition", Xiang Wang, Chenwei Wu, Jason D. Lee, Tengyu Ma, Rong Ge, NeurIPS 2020
>
> [4] "Fast and Guaranteed Tensor Decomposition via Sketching", Yining Wang, Hsiao-Yu Tung, Alexander Smola, Anima Anandkumar, NeurIPS 2015
>
> [5] "Online and Differentially-Private Tensor Decomposition", Yining Wang, Animashree Anandkumar, NeurIPS 2016
>
> [6] "Tensor Decompositions for Learning Latent Variable Models", Animashree Anandkumar, Rong Ge, Daniel Hsu, Sham M. Kakade, Matus Telgarsky, JMLR 2014

---

### Official Review · Reviewer_Tyn8 · 2025-11-01

**Soundness:** 2
**Presentation:** 3
**Contribution:** 2
**Rating:** 4
**Confidence:** 3

**Summary:**

In this paper, the authors propose a tensor power method (TPM) for CP decomposition of arbitrary-order tensors. Theoretical analysis demonstrates that the proposed method can achieve a running time of $\mathcal{O}(n^{p-1})$ per iteration on a tensor of order
$p$ and dimension $n$, overcoming limitations of the previous method (Wang & Anandkumar, 2016) restricted to lower-order tensors ($p\leq 3$). Theoretical analysis is provided for arbitrary including convergence guarantees and error bounds under bounded noise.

**Strengths:**

1. A new TPM method is proposed that can deal with CP decomposition of arbitrary-order tensors.
2. Theoretical analysis is given to guarantee the recovery, robustness to noise, and the advantage in time complexity per iteration.

**Weaknesses:**

1. There are no experiments that verify the advantage (efficiency and robustness) of the proposed method.
2. It is not clear what the difficulty is that extends the analysis in (Wang & Anandkumar, 2016) to the case with $p>3$.

**Questions:**

1. Please clarify what the difficulty is that extends the analysis in (Wang & Anandkumar, 2016) to the case with $p>3$.
2. As the author states that the proposed method can handle higher-order tensors with lower complexity, no experimental results are given to support the claims.
3. In the technique overview and conclusion, the author said that the proposed method can be applied to tensor data such as images and videos. Therefore, the author should carry out experiments on real-world image/video data along with the running time comparison to show the efficiency and robustness of the proposed method.

---

> ### Author Response · Authors · 2025-11-20
> **First response to reviewer Tyn8**
>
> ### We are grateful for your supportive evaluation and the recognition you’ve given to our efforts.
>
> ### Weakness 1 & Question 2, 3
>
> Thank you for your question regarding the empirical validation of our algorithms. While empirical validation would be interesting, demonstrating these guarantees experimentally would require substantial engineering effort and domain‑specific datasets that lie outside the paper’s scope, since our primary contribution is intentionally theoretical. We focused on rigorous analysis and matching upper/lower‑bound proofs.
>
> We would like to clarify that our work is purely a theoretical study in the tensor power method, which has a well-established place in top venues. Our paper is aligned with theoretical studies such as [1, 2, 3, 4], focusing on the theoretical aspects of tensor power method and optimization.
>
> ### Weakness 2 & Question 1
> We sincerely appreciate your continued engagement and thoughtful critiques. We would like to clarify the novelty and contributions of our work respectfully.
>
> #### Generalizing from $p \leq 3$ to arbitrary $p$.
> We agree that the properties of eigenvectors are generalized from [4], from bounded order ($p \leq 3$) to general order $p$, the proof requires a much different analysis and sketching technique.
>
> Our analysis derives p-order bounds on $\tan \theta(v_1, u_{t+1})$, and on the $\frac{\lambda_1|v_1^\top u_{t+1}|^{p-2}}{\lambda_j|v_j^\top u_{t+1}|^{p-2}} \ge 4\cdot 2^{p-2}$, and a more comprehensive analysis of the term $B_4 := \frac{|v_j^\top \widetilde{E}_{u_t}|}{\lambda_1|v_1^\top u_t|^{p-1}}$.We upper bound higher-order new properties that hold for all $p \geq 3$, which is not covered by the original proof in [4].
>
> #### Sketching techniques for faster algorithm
> Meanwhile, we provide a faster algorithm via non-trivial sketching techniques for the problem of tensor decomposition in [4], which achieves time complexity $\widetilde{O}(\epsilon^{-2}n^{p-1} + n^2k)$, compared to the original time of $O(nkdLR)$, where $R = \Omega(\log d + \log \log(\lambda_{\max}/\epsilon)), L = \Omega(\max\{K_0,k\}\log(\max\{K_0,k\}))$.
>
>
>
> ### Reference
>
> [1] "Guarantees for Alternating Least Squares in Overparameterized Tensor Decomposition", Dionysis Arvanitakis, Vaidehi Srinivas, Aravindan Vijayaraghavan, NeurIPS 2025
>
> [2] "Lower and Upper Bounds on the Pseudo-Dimension of Tensor Network Models
> ", Behnoush Khavari, Guillaume Rabusseau, NeurIPS 2021
>
> [3] "Beyond Lazy Training for Over-parameterized Tensor Decomposition", Xiang Wang, Chenwei Wu, Jason D. Lee, Tengyu Ma, Rong Ge, NeurIPS 2020
>
> [4] "Fast and Guaranteed Tensor Decomposition via Sketching", Yining Wang, Hsiao-Yu Tung, Alexander Smola, Anima Anandkumar, NeurIPS 2015
>
> [5] "Online and Differentially-Private Tensor Decomposition", Yining Wang, Animashree Anandkumar, NeurIPS 2016

---

### Official Review · Reviewer_c8NC · 2025-11-01

**Soundness:** 3
**Presentation:** 2
**Contribution:** 3
**Rating:** 6
**Confidence:** 5

**Summary:**

The paper studies a power method for computing low-rank CP decompositions of higher-order tensors.  It works with arbitrary tensor orders, unlike some previous related works.  The authors use randomized sketching to accelerate the tensor power method.  They provide thorough convergence analysis for perturbations of orthogonally decomposable tensors.

**Strengths:**

- The paper accelerates the tensor method power with randomized sketching to evaluate inner products more quickly.

- The paper has a strong theoretical contribution, with its Theorem 4.9.

- The work applies to tensors of any order.

**Weaknesses:**

- The paper is missing important references.

- The paper has no numerical experiments.

- The presentation in the paper is suboptimal, and the main body pushes too many key points to the appendices.

**Questions:**

1) General remark: The authors missed a very relevant line of work in the literature:
- "Landscape analysis of an improved power method for tensor decomposition", Joe Kileel, Timo Klock, Joao Pereira, NeurIPS 2021
- "Subspace power method for symmetric tensor decomposition", Joe Kileel, Joao Pereira, Numerical Algorithms, 2025.
There is also the recent preprint:
"Multi-subspace power method for decomposing all tensors", Kexin Wang, Joao Pereira, Joe Kileel, Anna Seigal, arXiv:2510.18627
This line of work develops a power method for CP decomposition that is applicable not only when the tensor is a noisy perturbation of an orthogonally decomposable tensor.  The method applies beyond the ranks where whitening is possible, and with tensors of any order.  Briefly, it works by constructing an auxiliary tensor $\tilde{T}$ to the input tensor $T$ such that the eigen/singular vectors of $\tilde{T}$ are the CP components of $T$.  The works also include an analysis of SS-HOPM and various noise robustness and convergence analyses.  I would suggest the authors cite these relevant works and comment very briefly on differences between their own work and these.

2) Line 170-171: As part of their stated breakthrough, the authors write ``Moreover, we have created a strong and adaptable algorithm that can handle a variety of tensor data: natural language corpora, images, videos, etc."  This adaptability to different data structures is something I didn't catch.  Can the authors explain more?

3) Line 267: It seems the authors are assuming the tensors $A_i$ have common CP components $x_j$.  This seems to be a strong assumption.  My guess is that the $A_i$'s are meant to be order-$(p-1)$ slices of a single low-rank order-$p$ tensor $A$ which would indeed yield this situation, but it wasn't explained in the "sketching technique" section.

3) Line 270: Can the authors comment on how the complexity improves over a non-sketching based implementation?  It seems the savings is through evaluation of $\langle A_i, u^{\otimes (p-1)} \rangle$ for $i=1, \ldots, n$ dropping from $\mathcal{O}(n^p)$ to $\mathcal{O}(\epsilon^{-2} n^{p-1})$.

4) Lines 294-311, Algorithm 1: This algorithm is not easily understood in a self-contained way.  I would suggest the authors include pseudo code for the subfunctions ds.QUERY and ds.QUERYVALUE within the body of the paper to increase readability.

5) Line 415: It is not necessary to include the fact that $(ab)^2 = a^2b^2$ for real numbers $a$ and $b$ in an ICLR submission.

6) Line 433, Theorem 4.9: Is $A^*$ orthogonally decomposable?  Please include this in the theorem statement if so.

7) General comment: The paper and appendices have no numerical experiments.  It would be nice to include some basic ones, for example to illustrate the speed-up brought by the sketching approach or the convergence rates proven by the authors.

8) General comment: The authors should be more explicit that they consider symmetric tensors and symmetric CP tensor decomposition.

---

> ### Author Response · Authors · 2025-11-20
> **First response to reviewer c8NC, part 1/2**
>
> ### We truly appreciate your positive comments and the time you invested in reviewing our work! We have addressed all your concerns below. We also made corresponding revisions in our latest pdf. All changes from the originally submitted version are highlighted in blue in the revised PDF.
>
> ### Weakness 1 & Question 1
>
> Thank you for pointing out these closely related works with important contributions, and we apologize for the oversight. In the revised version, we cite [1, 2, 3] in the related-work section and discuss their differences to our results.
>
> ### Weakness 2 & Question 8
>
> Thank you for your question regarding the empirical validation of our algorithms. While empirical validation would be interesting, demonstrating these guarantees experimentally would require substantial engineering effort and domain‑specific datasets that lie outside the paper’s scope, since our primary contribution is intentionally theoretical. We focused on rigorous analysis and matching upper/lower‑bound proofs.
>
> We would like to clarify that our work is purely a theoretical study in the tensor power method, which has a well-established place in top venues. Our paper is aligned with theoretical studies such as [4, 5, 6, 7], focusing on the theoretical aspects of tensor power method and optimization.
>
> ### Weakness 3
>
> Thank you for this remark on the presentation. Due to ICLR's strict page limit, we moved several related works and proof details to the appendix.
>
> ### Question 2
>
> We thank you for catching this and we agree that our wording was not clear enough. The “adaptability” we refer to is not that the algorithm directly reasons about sentences, pixels, or video frames, but our modality-agnostic algorithm only interacts with the input via the abstract tensor data structure. As long as natural language corpora, images, or videos are pre-processed into an $p$-th order tensor, the exact same algorithm and analysis apply without modification.
>
> ### Question 3
>
> Thank you for pointing out this lack of clarity. In the sketching section, we indeed work in the structured setting where there exists an underlying low-rank order-p tensor $A ^* = \sum_{i=1}^k \lambda_i v_i^{\otimes p}$, and the tensors $A_i$​ used in the sketching section are indeed order-$(p-1)$ slices. We revise the sketching section to explicitly introduce this construction.
>
> ### Question 4
>
> Our complexity, $\widetilde{O}(\epsilon^{-2}n^{p-1} + n^2k)$, is based on [8], with an additional $O(n^2 k)$ overhead (See the proof of lemma 4.2). The time complexity of query in [8] is $\widetilde{O}(\epsilon^{-2}(n+d)log1/\delta)$, which in our setting becomes $\widetilde{O}(\epsilon^{-2}n^{p-1})$.
>
> ### Question 5
>
> Thank you for the helpful suggestion. In the future, we will include pseudo code for ds.Query and ds.QueryValue.
>
> ### Question 6
>
> Thank you for this remark. In the revision, we remove this line and instead refer to this step with “by simple algebra”.
>
> ### Question 7
>
> We thank the reviewer for this helpful suggestion. Yes, throughout our analysis A^* is assumed to be orthogonally decomposable. We agree that Theorem 4.9 should be self-contained. In the revision, we state in the theorem statement that $A ^* = \sum_{i=1}^k \lambda_i v_i^{\otimes p}$ is orthogonal decomposable.
>
> ### Question 9
>
> Thank you for this comment. According to [9] and [10], the results can be directly extended to asymmetric tensors since they can first be symmetrized using simple matrix operations. For this reason, we did not emphasize the symmetry requirement in the current version. In the revision, we add that our analysis is presented for symmetric tensors and symmetric CP decomposition and note the extension to the asymmetric case.

---

> ### Author Response · Authors · 2025-11-25
> **First response to reviewer c8NC, part 2/2**
>
> ### Reference
>
> [1] "Landscape analysis of an improved power method for tensor decomposition", Joe Kileel, Timo Klock, Joao Pereira, NeurIPS 2021
>
> [2] "Subspace power method for symmetric tensor decomposition", Joe Kileel, Joao Pereira, Numerical Algorithms, 2025
>
> [3]  "Multi-subspace power method for decomposing all tensors", Kexin Wang, Joao Pereira, Joe Kileel, Anna Seigal, arXiv:2510.18627
>
> [4] "Guarantees for Alternating Least Squares in Overparameterized Tensor Decomposition", Dionysis Arvanitakis, Vaidehi Srinivas, Aravindan Vijayaraghavan, NeurIPS 2025
>
> [5] "Lower and Upper Bounds on the Pseudo-Dimension of Tensor Network Models
> ", Behnoush Khavari, Guillaume Rabusseau, NeurIPS 2021
>
> [6] "Beyond Lazy Training for Over-parameterized Tensor Decomposition", Xiang Wang, Chenwei Wu, Jason D. Lee, Tengyu Ma, Rong Ge, NeurIPS 2020
>
> [7] "Fast and Guaranteed Tensor Decomposition via Sketching", Yining Wang, Hsiao-Yu Tung, Alexander Smola, Anima Anandkumar, NeurIPS 2015
>
> [8] "Uniform Approximations for Randomized Hadamard Transforms with Applications", Yeshwanth Cherapanamjeri, Jelani Nelson, STOC 2022
>
> [9] "Online and Differentially-Private Tensor Decomposition", Yining Wang, Animashree Anandkumar, NeurIPS 2016
>
> [10] "Tensor Decompositions for Learning Latent Variable Models", Animashree Anandkumar, Rong Ge, Daniel Hsu, Sham M. Kakade, Matus Telgarsky, JMLR 2014

---

### Meta-Review · Area_Chair_tagj · 2026-01-04

**Summary:**

This paper presents a theoretical study of tensor power methods for computing the eigenvectors and eigenvalues of tensors. The flavor of paper itself is reflected in numerical linear algebra (NLA). All reviewers acknowledge the theoretical contributions, but would like to see empirical validation, even in synthetic data. The authors seem a little bit reluctant to produce numerical experiments, which I think is against the spirit of NLA and the ICLR community. In summary, the current theoretical findings might be far from the ICLR community if they cannot be empirically verified.

**Reviewer Concerns:**

All reviewers requested empirical validation, but the rebuttal did not address it.

**Reviewer Scores:**

I appreciate the authors' efforts to address the concerns raised in the rebuttal, and I don't think reviewers who asked for empirical validation would increase their score.

---

### Decision · Program_Chairs · 2026-01-26

Reject